

# Automatic differentiation of uncertainties: an interval computational differentiation for first and higher derivatives with implementation

Hend Dawood and Nefertiti Megahed

Department of Mathematics, Faculty of Science, Cairo University, Giza, Egypt

## ABSTRACT

Acquiring reliable knowledge amidst uncertainty is a topical issue of modern science. Interval mathematics has proved to be of central importance in coping with uncertainty and imprecision. Algorithmic differentiation, being superior to both numeric and symbolic differentiation, is nowadays one of the most celebrated techniques in the field of computational mathematics. In this connexion, laying out a concrete theory of interval differentiation arithmetic, combining subtlety of ordinary algorithmic differentiation with power and reliability of interval mathematics, can extend real differentiation arithmetic so markedly both in method and objective, and can so far surpass it in power as well as applicability. This article is intended to lay out a systematic theory of dyadic interval differentiation numbers that wholly addresses first and higher order automatic derivatives under uncertainty. We begin by axiomatizing a differential interval algebra and then we present the notion of an interval extension of a family of real functions, together with some analytic notions of interval functions. Next, we put forward an axiomatic theory of interval differentiation arithmetic, as a two-sorted extension of the theory of a differential interval algebra, and provide the proofs for its categoricity and consistency. Thereupon, we investigate the ensuing structure and show that it constitutes a multiplicatively non-associative S-semiring in which multiplication is subalternative and flexible. Finally, we show how to computationally realize interval automatic differentiation. Many examples are given, illustrating automatic differentiation of interval functions and families of real functions.

**Dedication.** In memory of Ramon Edgar Moore (1929–2015), the man who intervalized uncertainty.

# INTRODUCTION

Uncertainty arises in all fields of modern science. It is a state of limited knowledge where "To know" means "To be uncertain of". Acquiring reliable knowledge amidst uncertainty is the *raison d'être* of the present work. Motivated by an ever-increasing indeterminacy

Corresponding author
Hend Dawood,
hend.dawood@sci.cu.edu.eg

and complexity in physics and engineering and fueled by developments in computational and uncertainty mathematics, this work puts forward a categorical system of interval differentiation arithmetic that wholly addresses the computation of first and higher order automatic derivatives under uncertainty. Although scientists are fond of determinism, contemporary physical sciences have shown clearly that complete certainty is not reachable. The description of processes and states of physical systems discloses increasingly growing manipulations of uncertain quantifiable properties. Many features of the object world are rendered as numerical values that can either be measured or estimated by experts. Due to imperfection of our measuring methods, finiteness of our computations and lack of information, measured or estimated quantities can only be represented by finite approximations and thus are merely imprecise abstractions of reality (*Dawood & Dawood, 2019a*; *Dawood & Dawood, 2020*, and *Dawood & Dawood, 2022*).

In the effort to deal with the challenge of uncertainty, the subject of uncertainty mathematics has been developed in an extensive manner and many theoretical approaches have been introduced including fuzzy, probabilistic, and interval methods. A hot and fundamental topic of research that shades of into all approaches of uncertainty mathematics is interval analysis (see, *e.g.*, *Dawood, 2014*; *Dawood & Dawood, 2019a*, and *Dawood & Dawood, 2020*). The key advantage of the interval methods is that they provide "guaranteed interval enclosures" of the exact values of quantifiable uncertainties. In practice, when modelling physical systems, we have two distinct approaches: *getting guaranteed bounds* of an uncertain quantity and *computing a numerical approximation* thereof. The two approaches are not equivalent: the former includes the latter, but the latter does not imply the former. For example, to guarantee stability under uncertainty in control systems and robotics, it is crucial to compute guaranteed enclosures of the quantifiable features of the system under consideration (*Dawood, 2014* and *Dawood & Dawood, 2020*). Interval arithmetic brings forth a reliable way to cope with such problems. An interval number (a closed and bounded interval of real numbers) is a *guaranteed enclosure* of an imprecisely measured real-valued quantity, and an interval-valued function is consequently a guaranteed enclosure of a real-valued function under imprecision or uncertainty (or more generally, as we will see in this article, a reliable enclosure of the image of a *family* of real-valued functions). Historically speaking, the terms "interval arithmetic", "interval analysis", and "interval computations" are reasonably recent: they date from the fifties of the twentieth century. But the idea has been known since the third century BC, when Archimedes (287–212 BC) used lower and upper error bounds in the course of his computation of the constant $\pi$ (*Heath, 2009*). In the dawning of the twentieth century, the first rigorous developments of the theory of intervals appeared in the works of Norbert Wiener, John Charles Burkill, Rosalind Cecily Young, and Mihailo Petrovic (see *Wiener, 1921*, *Burkill, 1924*; *Young, 1931*; *Petrovic, 1932*, and *Petkovic, 2020*). Later, several distinguished developments of interval arithmetic appeared in the works of Paul S. Dwyer, Mieczyslaw Jan Warmus, Teruo Sunaga, and others (see, *e.g.*, *Dwyer, 1951*; *Warmus, 1956*, and *Sunaga, 1958*). However, it was not until 1959 that "interval analysis" in its modern sense was presented by the American mathematician and computer scientist, Ramon Edgar Moore (1929–2015), who was the first to recognize the power

of interval arithmetic as a viable computational apparatus for coping with uncertainty and imprecision (*Moore, 1959*). Nowadays, interval mathematics is a bold enterprise that comprises many different kinds of problem and has many fruitful applications in diverse areas of science and engineering (see, *e.g.*, *Allahviranloo, Pedrycz & Esfandiari, 2022*; *Beutner, Ong & Zaiser, 2022*, *Dawood, 2019*; *Dawood & Dawood, 2020*; *Dawood & Dawood, 2022*, *IEEE 1788 Committee, 2018*; *Jiang, Han & Xie, 2021*; *Kearfott, 2021*, *Mahato, Rout & Chakraverty, 2020*; *Matanga, Sun & Wang, 2022*, *Shary & Moradi, 2021*, and *Zheng et al., 2020*).

Two strands of research have led to the birth of the present work. The first strand starts from research in interval mathematics. The other strand stems from ordinary (real) automatic differentiation. Derivatives play an indispensable role in scientific computing. The expressions 'automatic differentiation', 'auto-differentiation', 'computational differentiation', 'algorithmic differentiation', and 'differentiation arithmetic' are in the just acceptation synonyms. They refer to a subtle and central tool to automatize the *simultaneous* computation of the numerical values of arbitrarily complex functions and their derivatives with no need for the symbolic representation of the derivative, only the function rule or an algorithm thereof is required (*Dawood & Megahed, 2019*). Auto-differentiation is thus *neither numeric nor symbolic*, nor is it a combination of both. It is also preferable to ordinary numerical methods: In contrast to the more traditional numerical methods based on finite differences, auto-differentiation is 'in theory' exact, and in comparison to symbolic algorithms, it is computationally inexpensive (*Dawood & Megahed, 2019*). The literature on algorithmic differentiation is immense and very diversified. For further reading, (see, *e.g.*, *Corliss & Rall, 1996*; *Dawood, 2014*, *Dawood & Megahed, 2019*; *Griewank & Walther, 2008*; *Moore, 1979*, *Neidinger, 2010*, and *Mitchell, 1991*). Currently, for its efficiency and accuracy in computing first and higher order derivatives, auto-differentiation is a celebrated technique with diverse applications in scientific computing and mathematics. It should therefore come as no surprise that there are numerous computational implementations of auto-differentiation. Among these, we mention, without pretension to be complete, INTLAB, Sollya, and InCLosure (see, *e.g.*, *Rump, 1999*, *Chevillard, Joldes & Lauter, 2010*, and *Dawood, 2020*). In practice, there are two *types* (*modes*) of algorithmic differentiation: a *forward-type* and a *reversed-type* (*Dawood & Megahed, 2019*). Presently, the two types are highly correlated and complementary and both have a wide variety of applications in, *e.g.*, non-linear optimization, sensitivity analysis, robotics, machine learning, computer graphics, and computer vision (see, *e.g.*, *Abdelhafez, Schuster & Koch, 2019*; *Dawood, 2014*, *Dawood & Megahed, 2019*; *Fries, 2019*, *Sommer, Pradalier & Furgale, 2016*, and *Tingelstad & Egeland, 2017*).

The use of ordinary auto-differentiation in the description and modeling of real world physical systems faces the problem of uncertainty. With the aid of interval mathematics, auto-differentiation can be intervalized to handle uncertainty in quantifiable properties of real world physical systems and accordingly provide the computational methods that suffice to deal with the important problem of "getting guaranteed bounds". Interval differentiation arithmetic combines subtlety of ordinary algorithmic differentiation with power and reliability of interval mathematics. By integrating the complementary

perspectives of both fields, interval differentiation arithmetic extends real differentiation arithmetic so markedly both in method and objective, and so far surpasses it in power and applicability. Real differentiation arithmetic, on the one hand, is concerned with the *simultaneous* calculation of the values of real functions and their derivatives with no requirement of the *symbolic* representation of the derivative. On the other hand, the subject matter of interval differentiation arithmetic is "interval functions" and its objective is the concurrent computation of *guaranteed enclosures* of images of real functions and their derivatives. This integration of interval and differentiation arithmetic is readily applicable to modelling and predicting the behaviour of real-world systems under uncertainty. Also, it has proved accuracy and efficiency in many scientific computations. As examples, we can mention enclosures of Taylor's coefficients, gradients, integrals, bounding boxes in ray tracing, and solutions of ordinary differential equations.

Three main problems have motivated the research conducted in this article. In the first place, despite its major importance in both basic research and practical applications, to the best of our knowledge, the algebraic aspects of interval differentiation arithmetic are not in-depth investigated. In the second place, almost no attempt has been made so far to explicitly axiomatize the theory of interval differentiation arithmetic in terms of clear and distinct elementary logical notions. In the third place, although an interval function is naturally an extension of a *family* of real functions, to the best of our knowledge, in all interval literature, the notion of a family of functions is not considered, and an interval function is assumed to extend a single real function. This presumption introduces an *unnecessary* restriction to the semantic of an interval function in the general sense. Families of functions arise naturally in many real-life and physical applications. In economics, the Cobb–Douglas family of production functions is an example; in physics, electron models, dynamical systems, quantum models, Camassa–Holm and Novikov wave-breaking models, and many other physical phenomena are described by families of functions (see, *e.g.*, *Cobb & Douglas, 1928*; *Silberberg & Suen, 2001*, *Anco, da Silva & Freire, 2015*, and *Engesser, Gabbay & Lehmann, 2011*). Providing the mathematical tools to get guaranteed enclosures of the images of families of real functions and their derivatives would provide an efficient way of predicting and controlling such physical systems and, thus, could have a substantial impact not only on theoretical research but also on many areas of applications. By the pursuit of this, formalizing the notion of a family of functions within the context of interval mathematics and interval differentiation arithmetic is one of the main motivations of this research.

Throughout the present text, we will understand by "interval differentiation arithmetic" ("interval differentiation algebra", "$\Delta_{\mathcal{J}}$-algebra", or "$\Delta_{\mathcal{J}}$-arithmetic") the fundamental algebraic structure underlying interval auto-differentiation as it is currently practised and implemented. It is our object, in this article, to present a consistent and categorical formalization of a theory of dyadic interval differentiation numbers ($\Delta_{\mathcal{J}}$-numbers) that fully addresses first and higher order auto-derivatives of families of real functions. The fundamental significance of categoricity is that if an axiomatization of $\Delta_{\mathcal{J}}$-numbers is categorical, then it correctly accounts, up to isomorphism, for *every structure* of $\Delta_{\mathcal{J}}$-arithmetic. The notion of categoricity is a bedrock of contemporary mathematics.

This is clearly described by John Corcoran in *Corcoran (1980)* and best reiterated in the words of Stewart Shapiro, "a categorical axiomatization is the best one can do" (*Shapiro, 1985*). In accordance with this *categorical* sense, the present article attempts to provide this "best" characterization. For this goal to be accomplished, we need to take a closer look at and formalize several fundamental analytic and algebraic concepts in the language of the theory to be axiomatized, so that one can establish the *metatheoretic* assertions of consistency and categoricity. This reformalization is mainly done in 'On theories and structures: some metatheoretical fundamentals' and 'A differential interval algebra'. In 'On theories and structures: some metatheoretical fundamentals', we set the stage by establishing the mathematical terminology, notions, and definitions that will be used throughout the rest of this article. 'Real differentiation arithmetic' is devoted to describing briefly the basic elements of the theory $\mathsf{T}_{\Delta_{\mathbb{R}}}$ of real differentiation arithmetic ($\Delta_{\mathbb{R}}$-arithmetic). In 'A differential interval algebra', we lay out an axiom system for the theory $\mathsf{T}_{\delta_{\mathcal{J}}}$ of a differential interval algebra and then we present the notion of an interval extension of a family of real functions, together with some analytic notions of interval functions. In 'A categorical axiomatization of interval differentiation arithmetic', we axiomatize a theory $\mathsf{T}_{\Delta_{\mathcal{J}}}$ of interval differentiation numbers ($\Delta_{\mathcal{J}}$-numbers) as a two-sorted extension of the theory $\mathsf{T}_{\delta_{\mathcal{J}}}$ of a differential interval algebra, and then we prove its consistency and categoricity. In order for the theory $\mathsf{T}_{\Delta_{\mathcal{J}}}$ to fully address and compute higher order and partial auto-derivatives using only *dyadic* $\Delta_{\mathcal{J}}$-numbers, in 'Differentiation extension of interval functions and higher-order auto-differentiability', we introduce the notion of a *differentiation extension* of interval functions, characterize *differentiability* for $\Delta_{\mathcal{J}}$-numbers, and establish their *differentiability conditions*. In 'The algebraic structure of interval differentiation arithmetic', we investigate the algebraic structure of $\Delta_{\mathcal{J}}$-arithmetic, establish its fundamental algebraic properties, and show that it forms a multiplicatively non-associative S-semiring in which multiplication is subalternative and flexible. Then, in 'Monotonicity and isomorphism theorems for interval differentiation numbers', we establish some monotonicity and isomorphism theorems for $\Delta_{\mathcal{J}}$-numbers and prove a result concerning the structure of $\Delta_{\mathbb{R}}$-numbers. Finally, in 'Machine implementation of interval auto-differentiation', we demonstrate the computational implementation of interval auto-differentiation and illustrate, by many numerical examples, how to concurrently compute guaranteed enclosures of images of both families of real functions and their first and higher order derivatives. The algorithms discussed in 'Machine implementation of interval auto-differentiation' are coded into reliable Common Lisp as a part of the software package, InCLosure[1] (*Dawood, 2020*). The InCLosure commands to calculate the results of the numerical examples are described and InCLosure input and output files are accessible as a supplementary material to the article (see *Dawood, 2020* and *Dawood, 2023*).

The attempted contribution of this article is therefore both a "logico-algebraic formalization" and an "extension" of interval differentiation arithmetic. The article gives an axiomatization of a comprehensive algebraic theory of interval differentiation arithmetic. Being based on clear and distinct elementary ideas of real and interval algebras, this formalized theory places the diverse approaches of interval auto-differentiation on

[1] InCLosure(Interval enCLosure) is an environment and a language for reliable scientific computing. Latest version of the software package InCLosure is freely accessible from CERN's archive *via* https://doi.org/10.5281/zenodo.2702404.

**Peer**J Computer Science

a firm and unified mathematical basis. We extend this theory in two directions. On the one hand, to the best of our knowledge, in almost all computational differentiation literature, researchers tend to 'borrow' or 'reinvent' *Clifford's* and *Grassmann algebras*[2] as proposed algebraic characterizations respectively for first and higher-order algorithmic differentiation. Without resorting to defining any sort of *Grassmann* structures, our axiomatization of *dyadic* interval differentiation numbers extends to fully address interval auto-derivatives of *first* and *higher* order. On the other hand, from the very beginning, our axiomatic system includes the notion of an interval extension of a family of real functions and the differentiability criteria thereof. By virtue of introducing this notion, the theory is extended to provide the mathematical tools to get guaranteed enclosures of the images of families of real functions and their derivatives. Also noteworthy here is that with a few basic modifications, the categorical system $\mathsf{T}_{\Delta_{\mathcal{J}}}$ axiomatized in this text can be extended analogously to compute *fuzzy auto-derivatives*.

[2]*Grassmann algebras* (or *exterior algebras*) were introduced by Hermann Grassmann in *Grassmann (1844)*. *Clifford's algebras*, introduced in *Clifford (1873)* by William Clifford, are Grassmann algebras in one dimension. Exterior algebras have nowadays many applications (see, *e.g.*, *Colombaro, Font-Segura & Martinez, 2020*; *Dawood & Megahed, 2019*, and *Trindade, Pinto & Floquet, 2019*).

## ON THEORIES AND STRUCTURES: SOME METATHEORETICAL FUNDAMENTALS

To achieve a rigorous formalization of the mathematical theory of this work, a specific formalized language and a particular logical apparatus are therefore required to attain all the results from obvious and distinct elementary mathematical concepts. So before we begin our axiomatization of *interval auto-differentiation*, we need to take a closer look at and formalize several preliminary mathematical concepts. To this aim, this section establishes the mathematical terminology, notions, and definitions that will be used throughout the rest of this article.

To make this article self-contained, we start by rehearsing some set-theoretical definitions. Let $\mathcal{A}$ be a set and let $\mathcal{A}^n$ be its $n$-th Cartesian power. A set $\Re$ is an $n$-ary (finitary) relation on $\mathcal{A}$ iff $\Re \subseteq \mathcal{A}^n$ and $\Re$ is a binary relation from $\mathcal{A}^{n-1}$ to $\mathcal{A}$. Thus, for $\mathbf{s} = (s_1, \ldots, s_{n-1}) \in \mathcal{A}^{n-1}$ and $t \in \mathcal{A}$, an $n$-ary relation $\Re$ is characterized to be $\Re \subseteq \mathcal{A}^n = \{(\mathbf{s}, t) | \mathbf{s} \in \mathcal{A}^{n-1} \wedge t \in \mathcal{A}\}$. Accordingly, a finitary relation $\Re$ is a binary relation whose *domain*, *range*, *field*, and *converse* are characterized to be, respectively $\mathrm{dom}(\Re) = \{\mathbf{s} \in \mathcal{A}^{n-1} | (\exists t \in \mathcal{A})(\mathbf{s}\Re t)\}$, $\mathrm{ran}(\Re) = \{t \in \mathcal{A} | (\exists \mathbf{s} \in \mathcal{A}^{n-1})(\mathbf{s}\widehat{\Re}t)\}$, $\mathrm{fld}(\Re) = \mathrm{dom}(\Re) \cup \mathrm{ran}(\Re)$, and $\widehat{\Re} = \{(t, \mathbf{s}) \in \mathcal{A}^n | \mathbf{s}\Re t\}$. Obviously $t\widehat{\Re}\mathbf{s} \Leftrightarrow \mathbf{s}\Re t$ and $\widehat{\widehat{\Re}} = \Re$ (*Dawood & Dawood, 2019a* and *Dawood & Dawood, 2020*).

Two indispensable definitions are those of *images* and *preimages* of finitary relations (see *Dawood & Dawood, 2019a* and *Dawood & Dawood, 2020*).

**Definition 2.1 (Images of Finitary Relations *Dawood & Dawood, 2020*):** *For* $1 \leq k \leq n-1$, *let* $\Re$ *be an* $n$-ary relation on $\mathcal{A}$, *and for* $(\mathbf{s}, t) \in \Re$, *let* $\mathbf{s} = (s_1, \ldots, s_{n-1})$, *with each* $s_k$ *is restricted to vary on* $S_k \subset \mathcal{A}$, *that is,* $\mathbf{s}$ *is restricted to vary on* $\mathbf{S} \subset \mathcal{A}^{n-1}$. *Then, the image of* $\mathbf{S}$ *(or the image of the sets* $S_k$*) with respect to* $\Re$, *in symbols* $I_{\Re}$, *is characterized to be*

$$T = I_{\Re}(\mathbf{S}) = I_{\Re}(S_1, \ldots, S_{n-1})$$
$$= \{t \in \mathcal{A} | (\exists \mathbf{s} \in \mathbf{S})(\mathbf{s}\Re t)\}$$
$$= \{t \in \mathcal{A} | (\exists_{k=1}^{n-1} s_k \in S_k)((s_1, \ldots, s_{n-1})\Re t)\}.$$

*The preimage* $\mathbf{S}$ *of* $T$ *is characterized to be the image of* $T$ *with respect to the converse* $\widehat{\mathfrak{R}}$ *of* $\mathfrak{R}$. *In other words*

$$\mathbf{S} = \mathrm{I}_{\widehat{\mathfrak{R}}}(T) = \{\mathbf{s} \in \mathcal{A}^{n-1} | (\exists t \in T)(t\widehat{\mathfrak{R}}\mathbf{s})\}.$$

In consequence of the equivalence $t\widehat{\mathfrak{R}}\mathbf{s} \Leftrightarrow \mathbf{s}\mathfrak{R}t$, apparently $T = \mathrm{I}_{\mathfrak{R}}(\mathbf{S}) \Leftrightarrow \mathbf{S} = \mathrm{I}_{\widehat{\mathfrak{R}}}(T)$.

In this sense, a general characterization of an *n-ary* (*finitary*) *function* can be introduced (*Dawood & Dawood, 2019a*). A set $q$ is an *n*-ary function (a function of *n* variables) on a set $\mathcal{A}$ iff $q$ is an $(n+1)$-ary relation on $\mathcal{A}$, and $(\forall \mathbf{s} \in \mathcal{A}^n)(\forall t, w \in \mathcal{A})(\mathbf{s}qt \wedge \mathbf{s}qw \Rightarrow t = w)$. That is, an *n*-ary (finitary) function is an $(n+1)$-ary relation. Restricting ourselves to the particular case of functions, we can pass up the set-theoretical notation $\mathbf{s}qt$ in favor of the common notation $t = q(\mathbf{s})$. In accord with this formulation, the preceding definitions of *domain*, *range*, *field*, and *converse* also apply to finitary functions. We say that a function $q$ is invertible or has an inverse $q^{-1}$ iff the converse relation $\widehat{q}$ is also a function, in which case $q^{-1} = \widehat{q}$ (*Dawood & Dawood, 2020*). Hereon, functions will be denoted by the letters $q$, $u$, and $v$. With a few exceptions, from now onwards, we will usually consider only *unary* functions.

In order to achieve the overarching objective of this work, it is necessary first to take a closer look at several *metamathematical*[3] concepts. A *metalinguistic* characterization of a *formalized theory* (an *axiomatic theory*) can be given. An axiomatic theory $\mathfrak{T}$ is characterized by an object formal language $\mathfrak{L}$ and a finite set of axioms $\Lambda_{\mathfrak{T}}$ (see *Dawood & Dawood, 2020*). Given an object formal language $\mathfrak{L}$ and a finite set $\Lambda_{\mathfrak{T}}$ of axioms ($\mathfrak{L}$-sentences), let $\varphi$ denote an $\mathfrak{L}$-sentence and let $\vDash_{\mathfrak{L}}$ denote the semantic consequence relation. The axiomatic $\mathfrak{L}$-theory $\mathfrak{T}$ of the set $\Lambda_{\mathfrak{T}}$ is the closure of $\Lambda_{\mathfrak{T}}$ under $\vDash_{\mathfrak{L}}$, that is $\mathfrak{T} = \{\varphi \in \mathfrak{L} | \Lambda_{\mathfrak{T}} \vDash_{\mathfrak{L}} \varphi\}$ (*Dawood & Dawood, 2020*). Next, the metatheoretical notions of a *model*, *categoricity*, and *consistency* of an axiomatic $\mathfrak{L}$-theory are characterized (see *Dawood & Megahed, 2019*).

**Definition 2.2 (Model of a Theory):** *Let* $\mathfrak{A}$ *be a mathematical structure (interpretation).* $\mathfrak{A}$ *is said to be a model of an axiomatic* $\mathfrak{L}$-*theory* $\mathfrak{T}$, *in symbols* $\mathfrak{A} \vDash \mathfrak{T}$, *iff every formula* $\varphi$ *of* $\mathfrak{T}$ *is satisfied by* $\mathfrak{A}$. *That is*

$$\mathfrak{A} \vDash \mathfrak{T} \Leftrightarrow (\forall \varphi \in \mathfrak{T})(\mathfrak{A} \vDash \varphi).$$

**Definition 2.3 (Categoricity of a Theory):** *Let* $\mathfrak{A}$ *and* $\mathfrak{B}$ *be any two models of an axiomatic* $\mathfrak{L}$-*theory* $\mathfrak{T}$. $\mathfrak{T}$ *is said to be categorical, in symbols* $\mathrm{Cat}(\mathfrak{T})$, *iff* $\mathfrak{A}$ *and* $\mathfrak{B}$ *are isomorphic. That is*

$$\mathrm{Cat}(\mathfrak{T}) \Leftrightarrow (\forall \mathfrak{A})(\forall \mathfrak{B})(\mathfrak{A} \vDash \mathfrak{T} \wedge \mathfrak{B} \vDash \mathfrak{T} \Rightarrow \mathfrak{A} \simeq \mathfrak{B}).$$

**Definition 2.4 (Consistency of a Theory):** *An axiomatic* $\mathfrak{L}$-*theory* $\mathfrak{T}$ *is said to be consistent, in symbols* $\mathrm{Con}(\mathfrak{T})$, *iff there is a model* $\mathfrak{A}$ *that satisfies the sentences of* $\mathfrak{T}$. *That is,* $\mathrm{Con}(\mathfrak{T}) \Leftrightarrow (\exists \mathfrak{A})(\mathfrak{A} \vDash \mathfrak{T})$.

[3]Metamathematics ("epitheory", or "metatheory") is the study of formalized mathematical theories and languages, and the interpretations thereof. A metatheoretical investigation produces *metatheorems* about the *object* theory under consideration (see, *e.g.*, *Curry, 1977*; *Hunter, 1971*, and *Kleene, 1952*).

A model of an axiomatic $\mathcal{L}$-theory $\mathfrak{T}$ is a mathematical structure $\mathfrak{A} = \langle \mathcal{A}, \sigma^{\mathfrak{A}} \rangle$ that makes the $\mathcal{L}$-sentences of $\mathfrak{T}$ true. Particular mathematical structures are indispensable for the objective of this work. These are defined next (*Dawood & Dawood, 2019a* and *Dawood & Dawood, 2020*).

**Definition 2.5 (Ringoid *Dawood & Dawood, 2019a*):** *A ring-like structure (or a ringoid) is an algebraic structure $\mathfrak{A} = \langle \mathcal{A}; +_{\mathcal{A}}, \times_{\mathcal{A}} \rangle$ with $+_{\mathcal{A}}$ and $\times_{\mathcal{A}}$ are total binary operations on the universe set $\mathcal{A}$. The operations $+_{\mathcal{A}}$ and $\times_{\mathcal{A}}$ are called respectively the addition and multiplication operations of the ringoid $\mathfrak{A}$.*

**Definition 2.6 (S-Ringoid *Dawood & Dawood, 2020*):** *A subdistributive ringoid (or an S-ringoid) is a ringoid $\mathfrak{A} = \langle \mathcal{A}; +_{\mathcal{A}}, \times_{\mathcal{A}} \rangle$ that satisfies at least one of the following subdistributive properties.*
(i)  *$(\forall s, t, w \in \mathcal{A})(s \times_{\mathcal{A}} (t +_{\mathcal{A}} w) \subseteq s \times_{\mathcal{A}} t +_{\mathcal{A}} s \times_{\mathcal{A}} w)$,*
(ii)  *$(\forall s, t, w \in \mathcal{A})((t +_{\mathcal{A}} w) \times_{\mathcal{A}} s \subseteq t \times_{\mathcal{A}} s +_{\mathcal{A}} w \times_{\mathcal{A}} s)$.*
Properties (i) and (ii) in the previous definition are called respectively left and right S-distributivity (or subdistributivity) (*Dawood & Dawood, 2020*).

**Definition 2.7 (Semiring *Dawood & Dawood, 2019a*):** *A ringoid $\mathfrak{A} = \langle \mathcal{A}; +_{\mathcal{A}}, \times_{\mathcal{A}} \rangle$ is a semiring iff $\mathfrak{A}$ satisfies the following properties.*
(i)  *$\mathcal{A}$ with $+_{\mathcal{A}}$ forms a commutative monoid with $0_{\mathcal{A}}$ is an identity for $+_{\mathcal{A}}$,*
(ii)  *$\mathcal{A}$ with $\times_{\mathcal{A}}$ forms a monoid with $1_{\mathcal{A}}$ is an identity for $\times_{\mathcal{A}}$,*
(iii)  *$\times_{\mathcal{A}}$ is both left and right distributive over $+_{\mathcal{A}}$,*
(iv)  *$0_{\mathcal{A}}$ is an annihilating element for $\times_{\mathcal{A}}$.*
 *If $\times_{\mathcal{A}}$ is commutative, then $\mathfrak{A}$ is said to be a commutative semiring.*

**Definition 2.8 (S-Semiring *Dawood & Dawood, 2020*):** *A subdistributive semiring (or an S-semiring) is an S-ringoid $\mathfrak{A} = \langle \mathcal{A}; +_{\mathcal{A}}, \times_{\mathcal{A}} \rangle$ that satisfies criteria (i), (ii), and (iv) in definition 2.7. A commutative S-semiring is one whose multiplication is commutative.*

It is important here to point out that an S-semiring generalizes the notion of a *near-semiring*; a near-semiring is a structure satisfying the axioms of a semiring except that it is *either* left or right distributive (For further details on near-semirings and related concepts, the reader can refer to *Clay, 1992*; *Pilz, 1983*, and *van Hoorn & van Rootselaar, 1967*).

Lastly, we define two new algebraic structures.

**Definition 2.9 (NA Semiring):** *A ringoid $\mathfrak{A} = \langle \mathcal{A}; +_{\mathcal{A}}, \times_{\mathcal{A}} \rangle$ is said to be an additively non-associative semiring (in short, +-NA semiring) iff $\mathfrak{A}$ satisfies (ii), (iii), and (iv) in definition 2.7, and $\langle \mathcal{A}; +_{\mathcal{A}} \rangle$ is a non-associative commutative monoid with identity element $0_{\mathcal{A}}$. Similarly, $\mathfrak{A}$ is said to be a multiplicatively non-associative semiring (in short, $\times$-NA semiring) iff $\mathfrak{A}$ satisfies (i), (iii), and (iv) in definition 2.7, and $\langle \mathcal{A}; \times_{\mathcal{A}} \rangle$ is a non-associative monoid with identity element $1_{\mathcal{A}}$.*

**Definition 2.10 (NA S-Semiring):** *An S-ringoid $\mathfrak{A} = \langle \mathcal{A}; +_{\mathcal{A}}, \times_{\mathcal{A}} \rangle$ is said to be an additively non-associative S-semiring (in short, +-NA S-semiring) iff $\mathfrak{A}$ satisfies (ii) and (iv) in definition 2.7, and $\langle \mathcal{A}; +_{\mathcal{A}} \rangle$ is a non-associative commutative monoid with identity element $0_{\mathcal{A}}$. Similarly, $\mathfrak{A}$ is said to be a multiplicatively non-associative S-semiring (in*

*short, $\times$-NA $\mathsf{S}$-semiring) iff $\mathfrak{A}$ satisfies (i) and (iv) in definition 2.7, and $\langle \mathcal{A}; \times_\mathcal{A} \rangle$ is a non-associative monoid with identity element $1_\mathcal{A}$.*

It is clear that if multiplication is commutative in a NA $\mathsf{S}$-semiring, then it is both left and right subdistributive.

## REAL DIFFERENTIATION ARITHMETIC

Before setting forth the assertions of an axiomatic theory of interval differentiation arithmetic in the succeeding sections, we need to describe briefly the basic elements of the theory $\mathsf{T}_{\Delta_\mathbb{R}}$ of *real differentiation arithmetic* (henceforth $\Delta_\mathbb{R}$-*arithmetic*). For further details and other constructions of $\Delta_\mathbb{R}$-arithmetic, the reader may consult, *e.g.*, *Dawood, 2014*; *Dawood & Megahed, 2019*, *Beda et al., 1959*; *Wengert, 1964*; *Moore, 1979*, *Rall, 1981*, and *Corliss & Rall, 1996*.

We hereon use the letters $q$, $u$, and $v$ as function symbols, and the letters $s$, $t$, and $w$ as real variable symbols. Given a class $\sigma = \{+, \times; -, ^{-1}; 0, 1, \leq\}$ of descriptive (non-logical) signs, let $\mathsf{R} = \langle \mathbb{R}; \sigma^\mathsf{R} \rangle$ be the field of real numbers, $\mathbb{R}_{\langle 1 \rangle}$ be the set of *unary* real functions, and $\delta$ be the differential operator for elements of $\mathbb{R}_{\langle 1 \rangle}$. For a $q$ in $\mathbb{R}_{\langle 1 \rangle}$, we use the predicate $\mathrm{diff}(q, s_0)$ to mean that $q$ is differentiable at some $s_0 \in \mathbb{R}$. We understand by a *differential real field* a structure $\mathsf{R}_\delta = \langle \mathbb{R}; \sigma^\mathsf{R}; \delta \rangle$ constructed by equipping $\mathsf{R}$ with the operator $\delta$ and its basic axioms. It is natural to begin with the definition of a *real differentiation number* ($\Delta_\mathbb{R}$-*number*).

**Definition 3.1 (Real Differentiation Numbers):** *The set of all real differentiation numbers ($\Delta_\mathbb{R}$-numbers, or $\Delta_\mathbb{R}$-pairs), with respect to a constant $s_0 \in \mathbb{R}$, is defined to be*

$$\mathbf{U}_\mathbb{R} = \left\{ \mathsf{q} \in \mathbb{R}^2 \,|\, \left( \exists q \in \mathbb{R}_{\langle 1 \rangle} \right) \left( \mathsf{q} = \left( q(s_0), \delta^1 q(s_0) \right) \wedge s_0 \in \mathrm{dom}(q) \wedge \mathrm{diff}(q, s_0) \right) \right\}.$$

That is, a $\Delta_\mathbb{R}$-number is an ordered pair of real numbers. Let the letters $\mathsf{q}$, $\mathsf{u}$, and $\mathsf{v}$, or equivalently the pairs $(q, \delta q)_{s_0}$, $(u, \delta u)_{s_0}$, and $(v, \delta v)_{s_0}$, be variable symbols ranging over the elements of $\mathbf{U}_\mathbb{R}$. Also, let $\mathsf{a}$, $\mathsf{b}$, and $\mathsf{c}$, or equivalently $(a, 0_\mathbb{R})_{s_0}$, $(b, 0_\mathbb{R})_{s_0}$, and $(c, 0_\mathbb{R})_{s_0}$, designate constants of $\mathbf{U}_\mathbb{R}$. In particular, we use $1_{\mathbf{U}_\mathbb{R}}$ to denote the $\Delta_\mathbb{R}$-number $(1_\mathbb{R}, 0_\mathbb{R})_{s_0}$ and $0_{\mathbf{U}_\mathbb{R}}$ to denote the $\Delta_\mathbb{R}$-number $(0_\mathbb{R}, 0_\mathbb{R})_{s_0}$.

The theory $\mathsf{T}_{\Delta_\mathbb{R}}$ of a *real differentiation algebra* (or a $\Delta_\mathbb{R}$-*algebra*) can then be axiomatized as follows (*Dawood & Megahed, 2019*).

**Definition 3.2 (Theory of Real Differentiation Algebra):** *Let $(q, \delta q)_{s_0}$ and $(u, \delta u)_{s_0}$ be in $\mathbf{U}_\mathbb{R}$. A differentiation algebra over a differential real field $\mathsf{R}_\delta = \langle \mathbb{R}; \sigma^\mathsf{R}; \delta \rangle$, or a $\Delta_\mathbb{R}$-algebra, is a two-sorted structure $\mathfrak{U}_\mathbb{R} = \langle \mathbf{U}_\mathbb{R}; \mathbb{R}; \sigma^{\mathfrak{U}_\mathbb{R}} \rangle$. The theory $\mathsf{T}_{\Delta_\mathbb{R}}$ of $\mathfrak{U}_\mathbb{R}$ is the deductive closure of the axioms of $\mathsf{R}_\delta$ together with the following sentences.*

(DA1)  $\Delta_\mathbb{R}$-*equality*. $(q, \delta q)_{s_0} =_{\mathbf{U}_\mathbb{R}} (u, \delta u)_{s_0} \Leftrightarrow q(s_0) =_\mathbb{R} u(s_0) \wedge \delta q(s_0) =_\mathbb{R} \delta u(s_0)$,

(DA2)  $\Delta_\mathbb{R}$-*addition*. $(q, \delta q)_{s_0} +_{\mathbf{U}_\mathbb{R}} (u, \delta u)_{s_0} = (q +_\mathbb{R} u, \delta q +_\mathbb{R} \delta u)_{s_0}$,

(DA3)  $\Delta_\mathbb{R}$-*multiplication*. $(q, \delta q)_{s_0} \times_{\mathbf{U}_\mathbb{R}} (u, \delta u)_{s_0} = (q \times_\mathbb{R} u, \delta q \times_\mathbb{R} u +_\mathbb{R} q \times_\mathbb{R} \delta u)_{s_0}$,

(DA4)  $\Delta_\mathbb{R}$-*negation*. $-_{\mathbf{U}_\mathbb{R}} (q, \delta q)_{s_0} = (-_\mathbb{R} q, -_\mathbb{R} (\delta q))_{s_0}$,

(DA5)  $\Delta_\mathbb{R}$-*reciprocal*. $(q, \delta q)_{s_0}^{-1_{\mathbf{U}_\mathbb{R}}} = (q^{-1_\mathbb{R}}, -_\mathbb{R} q^{-2} \times_\mathbb{R} \delta q)_{s_0}$.

Subtraction and division are defined as usual in terms of the four basic $\Delta_{\mathbb{R}}$-operations. For an economic exposition, we assert statements (DA2)–(DA5) as axioms but it should be mentioned that they are derivable from simpler statements. Hereafter, where no confusion is likely, the subscripts " $\mathbf{U}$", " $\mathbb{R}$", and " $s_0$" will be omitted. Also, we will usually write the structure $\mathfrak{U}_{\mathbb{R}}$ as $\langle \mathbf{U}_{\mathbb{R}}; +_{\mathbf{U}_{\mathbb{R}}}, \times_{\mathbf{U}_{\mathbb{R}}}; 0_{\mathbf{U}_{\mathbb{R}}}, 1_{\mathbf{U}_{\mathbb{R}}} \rangle$, omitting the universe set $\mathbb{R}$.

Differentiable real functions can be extended to $\Delta_{\mathbb{R}}$-numbers *via an extension principle* (*Dawood & Megahed, 2019*). Let $u$ be a real function differentiable at $s_0 \in \mathbb{R}$, that is there is $\mathsf{u} = (u, \delta u)_{s_0} \in \mathbf{U}_{\mathbb{R}}$, and let $\mathcal{Q}$ be a function rule. If $\mathcal{Q}_{\mathbb{R}}(u)$ is differentiable at $s_0$, then the differentiation extension of $\mathcal{Q}_{\mathbb{R}}$ is defined to be $\mathcal{Q}_{\mathbf{U}_{\mathbb{R}}}(\mathsf{u}) = (\mathcal{Q}_{\mathbb{R}}(u), \delta \mathcal{Q}_{\mathbb{R}}(u))_{s_0}$. For example, replacing $\mathcal{Q}$ by the " $\mathsf{sine}$" function, one obtains the *trigonometric* $\Delta_{\mathbb{R}}$-function $\sin(u, \delta u)_{s_0} = (\sin(u), \delta \sin(u))_{s_0}$.

We will not discuss the algebraic properties of $\Delta_{\mathbb{R}}$-numbers further in the present section, for these will be considered later in 'Monotonicity and isomorphism theorems for interval differentiation numbers', in the general framework of the theory $\mathsf{T}_{\Delta_{\mathcal{J}}}$ of interval differentiation arithmetic.

## A DIFFERENTIAL INTERVAL ALGEBRA

In order to axiomatize a categorical system of interval differentiation arithmetic ($\Delta_{\mathcal{J}}$-arithmetic) in the next sections, we need to lay out an axiom system for the theory $\mathsf{T}_{\delta_{\mathcal{J}}}$ of a differential interval algebra. The intended model of the axiomatic system $\mathsf{T}_{\delta_{\mathcal{J}}}$ is the differential S-semiring $\langle \mathcal{J}_{\mathbb{R}}, \mathbb{R}; +_{\mathcal{J}}, \times_{\mathcal{J}}; \delta \rangle$, where $\mathcal{J}_{\mathbb{R}}$ is the set of real closed intervals (interval numbers, or $\mathcal{J}$-numbers) and $\delta$ is the differential operator for unary interval functions ($\mathcal{J}$-functions).

To be able to prove categoricity and consistency of $\Delta_{\mathcal{J}}$-arithmetic, the first step towards axiomatizing the theory $\mathsf{T}_{\delta_{\mathcal{J}}}$ necessitates dealing first with the notion of *differentiability* in a continuously ordered field in a *purely syntactic way* (leaving out any references to mathematical analysis or possible interpretations). For further details on the syntactic approaches to these notions, see, *e.g.*, *Dawood, 2012*; *Dawood & Dawood, 2020*, *Montague, Kalish & Mar, 1980*; *Robinson, 1951*, and *Tarski, 1983*. The theory $\mathsf{T}_{\mathcal{F}}$ of *continuously ordered fields* (*cofields*) is characterized in the following definition *Dawood & Megahed, 2019*.

**Definition 4.1 (Theory of a Cofield):** *Let* $\mathfrak{F} = \langle \mathcal{F}; +_{\mathcal{F}}, \times_{\mathcal{F}}; 0_{\mathcal{F}}, 1_{\mathcal{F}}; \leq_{\mathcal{F}} \rangle$ *be a totally ordered field. The theory* $\mathsf{T}_{\mathcal{F}}$ *of a cofield (or a continuously ordered field) is the theory of* $\mathfrak{F}$ *together with the following axiom of continuity*

$$(\text{ACO}) \quad (\forall A \subseteq \mathcal{F})(\forall B \subseteq \mathcal{F})\left( \begin{array}{l} (\forall s \in A)(\forall t \in B)(s <_{\mathcal{K}} t) \Rightarrow \\ (\exists w \in \mathcal{F})(\forall s \in A)(\forall t \in B)(s \neq w \wedge t \neq w \Rightarrow s <_{\mathcal{K}} w \wedge w <_{\mathcal{K}} t) \end{array} \right).$$

We designate by $\geq_{\mathcal{F}}$ the converse of the non-strict total order $\leq_{\mathcal{F}}$, and by " $-_{\mathcal{F}}$" and " $^{-1}_{\mathcal{F}}$" the unary $\mathcal{F}$-operations of *negation* and *reciprocal*, respectively. *Subtraction* and *division* are defined as customary. From now onwards, when the context is clear, we may drop the subscript " $\mathcal{F}$".

[4]Many mathematicians use the term "complete ordered field" as a synonymous substitute for "continuously ordered field". Following Alfred Tarski (see, *e.g.*, *Tarski, 1994*), we pass up the adjective 'complete' in favor of the word 'continuously'. We reserve the word 'complete' for different logical uses.

For $n \geq 1$, let $\mathcal{K}_{\langle n \rangle}$ designate the class of all $n$-ary $\mathcal{F}$-functions. Hereon, the letters $q$, $u$, and $v$ are used as variable symbols ranging over the elements of $\mathcal{K}_{\langle 1 \rangle}$ (unary $\mathcal{F}$-functions). The intended interpretation (model) of the theory $\mathsf{T}_{\mathcal{F}}$ corresponds the structure $\mathfrak{F}$ to the continuously[4] (complete) ordered field $\langle \mathbb{R}; +_{\mathbb{R}}, \times_{\mathbb{R}}; 0_{\mathbb{R}}, 1_{\mathbb{R}}; \leq_{\mathbb{R}} \rangle$ of real numbers and $\mathcal{F}_{\langle 1 \rangle}$ is interpreted by the set $\mathbb{R}_{\langle 1 \rangle}$ of unary $\mathbb{R}$-functions.

Toward formalizing a differential interval algebra, we first need to extend the theory $\mathsf{T}_{\mathcal{F}}$ of a cofield by axiomatizing some analytic concepts. Let $q \in \mathcal{F}_{\langle 1 \rangle}$, and let $s$ and $l$ be, respectively, an $\mathcal{F}$-variable symbol and an $\mathcal{F}$-constant symbol. The 'limit' operator of the function $q(s)$ with respect to $m$, denoted $\lim_{s \to m} q(s)$, is defined thus (*Dawood & Megahed, 2019*):

$$\lim_{s \to m} q(s) = M \Leftrightarrow (\forall \epsilon > 0)(\exists \alpha > 0)\big(\forall s \in \mathrm{dom}(q)\big)\big(0 < |s - m| < \alpha \Rightarrow |q(s) - M| < \epsilon\big),$$

where the one-place operation symbol $|\cdot|$, called an $\mathcal{F}$-absolute value (or $\mathcal{F}$-modulus), is defined by

$$(\forall t \in \mathcal{F})(\exists w \in \mathcal{F})\big(|t| = w \Leftrightarrow (0 \leq t \wedge w = t) \vee (\neg 0 \leq t \wedge w = -t)\big).$$

If there is no such $M \in \mathcal{F}$, then the limit of $q$ at $m$ is said to be nonexistent in $\mathcal{F}$. For an $\mathcal{F}$-constant symbol $s_0 \in \mathrm{dom}(q)$, the 'continuity' predicate is a binary predicate, $\mathrm{cont}(q, s_0)$, defined by

$$\mathrm{cont}(q, s_0) \Leftrightarrow q(s_0) \in \mathcal{F} \wedge \lim_{s \to s_0} q(s) = q(s_0).$$

If $\mathrm{cont}(q, s_0)$ is true, then $q$ is said to be continuous at $s_0$. We also say that $q$ is continuous on $S_0 \subseteq \mathcal{F}$ iff it is continuous at all $s_0 \in S_0$, that is

$$\mathrm{cont}(q, S_0) \Leftrightarrow (\forall s_0 \in S_0)\big(\mathrm{cont}(q, s_0)\big).$$

**Definition 4.2** (*n-Differential $\mathcal{F}$-Operator Dawood & Megahed, 2019*): *Let $n \geq 0$, and let $s$ and $\beta$ be $\mathcal{F}$-variable symbols. The $n$-differential $\mathcal{F}$-operator of a function $q(s) \in \mathcal{F}_{\langle 1 \rangle}$, denoted $\delta^n q(s)$, is characterized recursively by the following equations.*

(i) $\delta^0 q(s) = q(s)$,

(ii) $\delta^1 q(s) = \lim_{\beta \to 0_{\mathcal{F}}} \frac{q(s+\beta) - q(s)}{\beta} = \delta^1 \delta^0 q(s)$,

(iii) $n \geq 1 \Rightarrow \delta^n q(s) = \delta^1 \delta^{n-1} q(s)$.

Evidently if the limit in definition 4.2 exists, then the $n$-differential $\delta^n q(s)$ of $q$ is consequently a unary $\mathcal{F}$-function. Henceforth, we will usually write $\delta^n q$ and $\delta q$ for $\delta^n q(s)$ and $\delta^1 q(s)$, respectively.

Closely related to the differential operator is the $n$-differentiability predicate, which is characterized as follows.

**Definition 4.3** (*n-Differentiability $\mathcal{F}$-Predicate Dawood & Megahed, 2019*): *Let $n \geq 0$, let $q(s) \in \mathcal{F}_{\langle 1 \rangle}$, and let $s_0 \in \mathrm{dom}(q)$ be an $\mathcal{F}$-constant symbol. The ternary $n$-differentiability $\mathcal{F}$-predicate for the function $q$, denoted $\mathrm{diff}^n(q, s_0)$, is defined by*

$$\mathrm{diff}^n(q, s_0) \Leftrightarrow \delta^n q(s_0) \in \mathcal{F}.$$

*If $\mathrm{diff}^n(q, s_0)$ is true, then $q$ is said to be $n$-differentiable at $s_0$.*

[5]There are so many systems of interval algebras (see, *e.g.*, *Hansen, 1975*, *Kulisch, 2013*; *Gardenyes, Mielgo & Trepat, 1985*; *Markov, 1995*; *Kaucher, 1980*; *Shary, 2002*; *Piegat & Dobryakova, 2020*, *Dawood, 2011*; *Dawood, 2012*; *Dawood & Dawood, 2019b*, and *Dawood & Dawood, 2020*). Here we axiomatize classical (naive) interval algebra as introduced in, *e.g.*, *Alefeld & Mayer, 2000*; *Moore, 1979*, *Dawood, 2014*, and *Dawood, 2019*. An axiom system for an interval differentiation algebra over a different theory of intervals will be fundamentally the same as the axiom system presented in this text, but it might differ in the resulting algebraic structure.

Since for $s_0 \in \mathrm{dom}(q)$, $\delta^0 q(s_0) = q(s_0) \in \mathcal{F}$, the predicate $\mathrm{diff}^0(q, s_0)$ is always true and accordingly every $q \in \mathcal{F}_{\langle 1 \rangle}$ is 0-differentiable at $s_0 \in \mathrm{dom}(q)$. Apparently, if $\mathrm{diff}^n(q, s_0)$ is true, then for $0 \le m < n$, $\delta^m q(s_0) \in \mathcal{F}$.

**Definition 4.4 (Continuous Differentiability $\mathcal{F}$-Predicate ):** *Let $n \ge 0$, let $q(s) \in \mathcal{F}_{\langle 1 \rangle}$, and let $s_0 \in \mathrm{dom}(q)$ be an $\mathcal{F}$-constant symbol. The continuous $n$-differentiability $\mathcal{F}$-predicate for the function $q$, denoted $\mathrm{cdiff}^n(q, s_0)$, is characterized recursively by the following statements.*

(i)   $\mathrm{cdiff}^0(q, s_0) \Leftrightarrow \mathrm{cont}(q, s_0)$,

(ii)  $\mathrm{cdiff}^1(q, s_0) \Leftrightarrow \mathrm{cdiff}^0(q, s_0) \wedge \mathrm{cont}(\delta^1 q, s_0)$,

(iii) $n \ge 1 \Rightarrow \mathrm{cdiff}^n(q, s_0) \Leftrightarrow \mathrm{cdiff}^{n-1}(q, s_0) \wedge \mathrm{cont}(\delta^n q, s_0)$.

*If $\mathrm{cdiff}^n(q, s_0)$ is true, then $q$ is said to be continuously $n$-differentiable at $s_0$.*

In a manner analogous to the differential operator, if $\mathrm{cdiff}^n(q, s_0)$ is true, then for $0 \le m < n$, $\mathrm{cdiff}^m(q, s_0)$ is true as well.

A theory $\mathsf{T}_{\mathcal{J}}$ of an *interval algebra* or a *classical*[5] *interval algebra* (henceforth a $\mathcal{J}$-*algebra*) over a cofield can then be characterized as follows (*Dawood & Dawood, 2020* and *Dawood & Dawood, 2022*).

**Definition 4.5 (Theory of Interval Algebra):** *Let $\sigma = \{+, \times; -, ^{-1}; 0, 1\}$ be a class of descriptive (non-logical) signs, and let $\mathfrak{F} = \langle \mathcal{F}; \sigma^{\mathfrak{F}} \rangle$ be a cofield. The theory $\mathsf{T}_{\mathcal{J}}$ of an interval algebra (a $\mathcal{J}$-algebra) over $\mathfrak{F}$ is the theory of a many-sorted algebraic structure $\mathfrak{I}_{\mathcal{F}} = \langle \mathcal{J}_{\mathcal{F}}; \mathcal{F}; \sigma^{\mathfrak{I}_{\mathcal{F}}} \rangle$ axiomatized by the following sentences.*

(I1)  $(\forall S \in \mathcal{J}_{\mathcal{F}}) \left( S = \{s \in \mathcal{F} \mid (\exists \underline{s} \in \mathcal{F})(\exists \bar{s} \in \mathcal{F})(\underline{s} \le_{\mathcal{F}} s \le_{\mathcal{F}} \bar{s})\} \right)$,

(I2)  $(\forall S, T \in \mathcal{J}_{\mathcal{F}})(\circ \in \{+, \times\} \Rightarrow S \circ_{\mathcal{J}} T = \{w \in \mathcal{F} \mid (\exists s \in S)(\exists t \in T)(w = s \circ_{\mathcal{F}} t)\})$,

(I3)  $(\forall S \in \mathcal{J}_{\mathcal{F}})(\diamond \in \{-\} \vee (\diamond \in \{^{-1}\} \wedge 0_{\mathcal{J}} \nsubseteq S)) \Rightarrow \diamond_{\mathcal{J}} S = \{w \in \mathcal{F} \mid (\exists s \in S)(w = \diamond_{\mathcal{F}} s)\}$.

Axiom (I1) of the above definition characterizes what a $\mathcal{J}$-*number* (an interval number, or an $\mathcal{F}$-interval) is. Axioms (I2) and (I3) prescribe, respectively the binary operations of $\mathcal{J}$-*addition* (" $+_{\mathcal{J}}$") and $\mathcal{J}$-*multiplication* (" $\times_{\mathcal{J}}$"), and the unary operations of $\mathcal{J}$-*negation* (" $-_{\mathcal{J}}$") and $\mathcal{J}$-*reciprocal* (" $^{-1}{}_{\mathcal{J}}$"). The intended model of $\mathsf{T}_{\mathcal{J}}$ corresponds the sets "$\mathcal{F}$" and " $\mathcal{J}_{\mathcal{F}}$" to the sets " $\mathbb{R}$" and " $\mathcal{J}_{\mathbb{R}}$" (of real numbers and real closed intervals), respectively, and the symbols " $\circ_{\mathcal{F}}$", and " $\diamond_{\mathcal{F}}$" to the binary and unary $\mathbb{R}$-operations.

In the sequel, the upper-case letters $S$, $T$, and $W$, or equivalently $[\underline{s}, \bar{s}]$, $[\underline{t}, \bar{t}]$, and $[\underline{w}, \overline{w}]$, will be used as variable symbols ranging over the domain $\mathcal{J}_{\mathcal{F}}$ of $\mathcal{J}$-numbers. A *point (singleton,* or *degenerate) $\mathcal{J}$-numbers* $\{s\}$ will be denoted by $[s]$. Also, the letters $A$, $B$, and $C$, or equivalently $[\underline{a}, \bar{a}]$, $[\underline{b}, \bar{b}]$, and $[\underline{c}, \bar{c}]$, will be used to designate constants of $\mathcal{J}_{\mathcal{F}}$. In particular, we will use $1_{\mathcal{J}}$ and $0_{\mathcal{J}}$ to designate, respectively, the singleton $\mathcal{J}$-numbers $\{1_{\mathcal{F}}\}$ and $\{0_{\mathcal{F}}\}$. It is convenient here to single out the set $\mathcal{J}_{[s]}$ of *point $\mathcal{J}$-numbers*. This is defined thus:

$$\mathcal{J}_{[s]} = \{S \in \mathcal{J}_{\mathcal{F}} \mid (\exists s \in \mathcal{F})(S = [s, s])\}.$$

Equality of $\mathcal{J}$-numbers is an immediate consequence of the axiom of extensionality[6] of set theory plus the fact that a $\mathcal{J}$-number is a totally $\le$-ordered subset of $\mathcal{F}$. Precisely,

[6]For two sets $\mathcal{A}$ and $\mathcal{B}$, $\mathcal{A} = \mathcal{B} \Leftrightarrow (\forall w)(w \in \mathcal{A} \Leftrightarrow w \in \mathcal{B})$.

$$[\underline{s}, \bar{s}] =_{\mathcal{I}} [\underline{t}, \bar{t}] \Leftrightarrow \underline{s} =_{\mathcal{F}} \underline{t} \wedge \bar{s} =_{\mathcal{F}} \bar{t}.$$

The categoricity of the theory $\mathsf{T}_{\mathcal{J}}$ of $\mathcal{J}$-algebra is established by the following theorem.

**Theorem 4.1 (Categoricity of the Interval Theory):** *The theory* $\mathsf{T}_{\mathcal{J}}$ *of* $\mathcal{J}$*-algebra is categorical. That is,* $\mathrm{Cat}(\mathsf{T}_{\mathcal{J}})$.

**Proof** Let $\sigma = \{+, \times; -, ^{-1}; 0, 1\}$ be a class of descriptive (non-logical) signs of $\mathcal{L}$, and let $\mathfrak{J}_1 = \langle \mathcal{J}_1; \mathcal{F}_1; \sigma^{\mathfrak{J}_1} \rangle$ and $\mathfrak{J}_2 = \langle \mathcal{J}_2; \mathcal{F}_2; \sigma^{\mathfrak{J}_2} \rangle$ be two structures such that $\mathfrak{J}_1 \models \mathsf{T}_{\mathcal{J}} \wedge \mathfrak{J}_2 \models \mathsf{T}_{\mathcal{J}}$. Accordingly, $\langle \mathcal{F}_1; \sigma^{\mathcal{F}_1} \rangle$ and $\langle \mathcal{F}_2; \sigma^{\mathcal{F}_2} \rangle$ are two cofields. A theory of cofields is categorical, that is, there is one and up to isomorphism only one cofield. The structure $\langle \mathbb{R}; \sigma^{\mathbb{R}} \rangle$ is characterized, up to isomorphism, as the only cofield.

Let $i : \mathcal{F}_1 \hookrightarrow \mathcal{F}_2$ be the isomorphism from $\mathcal{F}_1$ onto $\mathcal{F}_2$. We can then define $I : \mathcal{J}_1 \hookrightarrow \mathcal{J}_2$ by

$$I(S) = I([\underline{s}, \overline{s}]) = [i(\underline{s}), i(\overline{s})],$$

for all $S = [\underline{s}, \overline{s}]$ in $\mathcal{J}_1$ where $\underline{s}, \overline{s} \in \mathcal{F}_1$. By definition 4.5, It is straightforward to show that $I$ is an isomorphism from $\mathcal{J}_1$ onto $\mathcal{J}_2$. This proves that $\mathsf{T}_{\mathcal{J}}$ is categorical. $\qquad\square$

That is, the theory $\mathsf{T}_{\mathcal{J}}$ uniquely characterizes the algebra of $\mathcal{J}$-numbers, and the structure $\langle \mathcal{J}_{\mathbb{R}}; \mathbb{R}; +_{\mathcal{J}}, \times_{\mathcal{J}}; 0_{\mathcal{J}}, 1_{\mathcal{J}} \rangle$ is, up to isomorphism, the only possible model of $\mathsf{T}_{\mathcal{J}}$. Accordingly, in establishing our assertions about $\mathcal{J}$-numbers, the properties of real numbers are assumed in advance.

By means of definition 4.5 and from the fact that $\mathcal{J}$-numbers are *ordered sets* of $\mathbb{R}$, the following theorem is derivable (*Dawood, 2012* and *Dawood & Dawood, 2020*).

**Theorem 4.2 (Interval Operations):** *Let* $[\underline{s}, \overline{s}]$ *and* $[\underline{t}, \overline{t}]$ *be two* $\mathcal{J}$*-numbers. The binary and unary* $\mathcal{J}$*-operations (interval operations) are formulated thus:*

(i) $\mathcal{J}$*-addition.* $[\underline{s}, \overline{s}] +_{\mathcal{J}} [\underline{t}, \overline{t}] = [\underline{s} +_{\mathbb{R}} \underline{t}, \overline{s} +_{\mathbb{R}} \overline{t}]$,

(ii) $\mathcal{J}$*-multiplication.* $[\underline{s}, \overline{s}] \times_{\mathcal{J}} [\underline{t}, \overline{t}] = [\min P, \max P]$,

(iii) $\mathcal{J}$*-negation.* $-_{\mathcal{J}} [\underline{s}, \overline{s}] = [-_{\mathbb{R}} \overline{s}, -_{\mathbb{R}} \underline{s}]$,

(iv) $\mathcal{J}$*-reciprocal.* $0_{\mathcal{J}} \not\subseteq [\underline{s}, \overline{s}] \Rightarrow [\underline{s}, \overline{s}]^{-1_{\mathcal{J}}} = [\overline{s}^{-1_{\mathbb{R}}}, \underline{s}^{-1_{\mathbb{R}}}]$,

*where* $P = \{\underline{s} \times_{\mathbb{R}} \underline{t}, \underline{s} \times_{\mathbb{R}} \overline{t}, \overline{s} \times_{\mathbb{R}} \underline{t}, \overline{s} \times_{\mathbb{R}} \overline{t}\}$, *and* $\min$ *and* $\max$ *are respectively the* $\leq_{\mathbb{R}}$*-minimal and* $\leq_{\mathbb{R}}$*-maximal.*

If no confusion is likely, we will often omit the subscripts $\mathcal{J}$ and $\mathbb{R}$. It is clear that interval addition, multiplication, and negation are *total* $\mathcal{J}$-operations, while interval reciprocal is a *partial* $\mathcal{J}$-operation. As customary, interval subtraction and division are defined respectively as $S - T = S + (-T)$ and $S \div T = S \times (T^{-1})$.

The set-theoretic characterization of interval arithmetic brings to the fore a peculiar feature that seems strange at first. Definition 4.5 entails that a $\mathcal{J}$-operation considers *all occurrences* of variables as *independent* (*Dawood & Dawood, 2020*). Let two $\mathcal{J}$-variables $S$ and $T$ be assigned the same $\mathcal{J}$-constant $[-1, 0]$. Evidently, $S \times_{\mathcal{J}} S = S \times_{\mathcal{J}} T = [-1, 1]$ which is equal to the image, $\mathrm{I}_{\mathrm{ind}}$, of the *multivariate* $\mathbb{R}$-function $q_{\mathrm{ind}}(s, t) = s \times_{\mathbb{R}} t$, with $s, t \in [-1, 0]$. Now consider a unary $\mathbb{R}$-function $q_{\mathrm{dep}}(s) = s \times_{\mathbb{R}} s$, with $s \in [-1, 0]$. The image $\mathrm{I}_{\mathrm{dep}}$ of $q_{\mathrm{dep}}$ is $[0, 1]$. Provided that images of $\mathbb{R}$-functions are *inclusion monotonic* (see, e.g., *Dawood, 2012* and *Dawood & Dawood, 2019b*), we have the nice enclosure $\mathrm{I}_{\mathrm{dep}} \subseteq \mathrm{I}_{\mathrm{ind}}$, and therefore the result of a $\mathcal{J}$-operation $S \times_{\mathcal{J}} T$ is a *guaranteed interval enclosure* of the

image of the corresponding $\mathbb{R}$-function. Although this is typically appraised as one of the strengths of interval analysis, in many practical situations, interval enclosures might be too wide to be beneficial. The name of this crucial phenomenon is the *interval dependency problem*, a concept that we make precise in the next theorem (see *Dawood & Dawood, 2019a* and *Dawood & Dawood, 2020*).

**Theorem 4.3 (Dependency Problem):** *Let $S_i$ be $\mathcal{J}$-numbers, for $1 \leq i \leq n$. Let $q_{\mathbb{R}}(s_1, \ldots, s_n)$ be a continuous $\mathbb{R}$-function with $s_i \in S_i$, and let $q_{\mathcal{J}}(S_1, \ldots, S_n)$ be a $\mathcal{J}$-function defined by the same rule as $q_{\mathbb{R}}$. The result of computing the image of the intervals $S_i$ under $q_{\mathbb{R}}$, denoted $\mathrm{I}_q(S_1, \ldots, S_n)$, using classical $\mathcal{J}$-operations (definition 4.2), cannot be generally exact if some $s_i$ are functionally dependent. That is,*

(i) $\quad (\exists q)\left(\mathrm{I}_q(S_1, \ldots, S_n) \neq q_{\mathcal{J}}(S_1, \ldots, S_n)\right)$.

*In general,*

(ii) $\quad (\forall q)\left(\mathrm{I}_q(S_1, \ldots, S_n) \subseteq q_{\mathcal{J}}(S_1, \ldots, S_n)\right)$.

What this theorem shows is that the result obtained by the $\mathcal{J}$-function $q_{\mathcal{J}}$ is usually *overestimated* due to the presence of functional dependence. Interval dependency is a 'deep-rooted' problem, dating back to the early works on interval arithmetic. A recent investigation of the logical underpinnings and some ways out of the problem can be found in *Dawood & Dawood (2019a)* and *Dawood & Dawood (2020)*. A plausible definition and a graphical representation (dependency diagrams) of the dependence of interval variables were also proposed in *Shary & Moradi (2021)*. Plenty of effort has been made to administer feasible remedies. With convenient refined techniques, the interval enclosure $q_{\mathcal{J}}$ can be made arbitrarily close to the image $\mathrm{I}_q$. By noting regions of monotonicity, one technique is defining the *elementary interval functions* as the *exact images* of their corresponding real counterparts. Let $n$ be a nonnegative integer and $S = [\underline{s}, \overline{s}]$ be a $\mathcal{J}$-number. We can define as instances

$$e^S = \left[e^{\underline{s}}, e^{\overline{s}}\right], \ln(S) = \left[\ln(\underline{s}), \ln(\overline{s})\right] \text{ if } \underline{s} > 0;$$

$$\sqrt{S} = \left[\sqrt{\underline{s}}, \sqrt{\overline{s}}\right] \text{ if } \underline{s} \geq 0, \sin(S) = \left[\min_{s \in S}(\sin(s)), \max_{s \in S}(\sin(s))\right];$$

$$S^n = \begin{cases} \left[\underline{s}^n, \overline{s}^n\right] & \text{iff } \underline{s} > 0 \text{ or } n \text{ is odd,} \\ \left[\overline{s}^n, \underline{s}^n\right] & \text{iff } \overline{s} < 0 \text{ and } n \text{ is even,} \\ \left[0, |S|^n\right] & \text{iff } 0 \in S \text{ and } n \text{ is even;} \end{cases}$$

where $|S| = \max\{|\underline{s}|, |\overline{s}|\}$ is the $\mathcal{J}$-absolute value (or $\mathcal{J}$-modulus) of $S$.

Performing *naive $\mathcal{J}$-arithmetic* (theorem 4.2) on these exact images we can get *sharper* enclosures of their algebraic combinations. Moreover, a diversity of interval methods has been devised to compute narrower interval enclosures. Without pretension to be complete, we can mention the *subdivision method, centered forms, circular complex centered forms, generalized centered forms, Hansen's method, remainder forms* (see, e.g., *Dawood & Dawood, 2019a*; *Moore, 1979*, *Rokne & Ratschek, 1984*, *Kulisch, 2013* and *Alefeld & Mayer, 2000*). For instance, the subdivision method presented by Moore in *Moore, 1966* and *Moore, 1979* is

a celebrated method that can be described as follows. Let $S = [\underline{s}, \bar{s}]$ be a $\mathcal{J}$-number. First, subdivision of $S$ into $n$ subintervals $S_i$ is applied such that

$$S_i = [\underline{s} + (i-1)\ell(S)/n, \underline{s} + (i)\ell(S)/n],$$

where $\ell(S) = \bar{s} - \underline{s}$ and $\ell(S_i) = \ell(S)/n$ are respectively the *widths* (*lengths*) of $S$ and $S_i$. Consequently $S = \cup_{i=1}^{n} S_i$. Then, evaluating a $\mathcal{J}$-function $q_{\mathcal{J}}$ for each subinterval $S_i$ yields the enclosure (*Dawood, 2014*)

$$\mathrm{I}_q(S) \subseteq \cup_{i=1}^{n} q_{\mathcal{J}}(S_i) \subseteq q_{\mathcal{J}}(S).$$

As the number $n$ of subintervals gets larger, $\cup_{i=1}^{n} q_{\mathcal{J}}(S_i)$ gets arbitrarily close to the exact image $\mathrm{I}_q(S)$. The subdivision method thus gives sharper enclosures than the naive evaluation $q_{\mathcal{J}}(S)$. In 'Machine implementation of interval auto-differentiation', we will deploy the subdivision method in order to compute reliable and realistic enclosures of families of real functions and their derivatives.

The characterization of the interval algebraic operations implies a number of familiar algebraic properties. However, being a particular kind of set arithmetic, interval arithmetic ($\mathcal{J}$-arithmetic) has certain peculiar properties involving set inclusion. The singleton intervals $0_{\mathcal{J}}$ and $1_{\mathcal{J}}$ are identities for $\mathcal{J}$-addition and $\mathcal{J}$-multiplication, respectively; $\mathcal{J}$-addition and $\mathcal{J}$-multiplication are both associative and commutative; $\mathcal{J}$-addition is cancellative; $\mathcal{J}$-multiplication is cancellative only for all $S \not\supseteq 0_{\mathcal{J}}$; a $\mathcal{J}$-number is invertible for $\mathcal{J}$-addition (respectively, $\mathcal{J}$-multiplication) if and only if it is a singleton $\mathcal{J}$-number (respectively, a nonzero singleton $\mathcal{J}$-number); and $\mathcal{J}$-multiplication left and right S-*distributes* over $\mathcal{J}$-addition (see definition 2.6 of 'On theories and structures: some metatheoretical fundamentals'). In other words, in accordance with definition 2.8, the structure $\langle \mathcal{J}_{\mathbb{R}}; +_{\mathcal{J}}, \times_{\mathcal{J}}; 0_{\mathcal{J}}, 1_{\mathcal{J}} \rangle$ of classical $\mathcal{J}$-numbers can be shown to be a *commutative* S-*semiring* (*Dawood & Dawood, 2019a* and *Dawood & Dawood, 2020*).

Throughout this text we will make use of the following theorems (see *Dawood, 2014* and *Dawood & Dawood, 2020*).

**Theorem 4.4 (Inclusion Monotonicity for $\mathcal{J}$-Numbers):** *Let $S_1$, $S_2$, $T_1$, and $T_2$ be $\mathcal{J}$-numbers such that $S_1 \subseteq T_1$ and $S_2 \subseteq T_2$. Let $\circ_{\mathcal{J}} \in \{+, \times\}$ be a binary $\mathcal{J}$-operation and $\diamond_{\mathcal{J}} \in \{-, ^{-1}\}$ be a definable unary $\mathcal{J}$-operation. Then*

(i)   $S_1 \circ_{\mathcal{J}} S_2 \subseteq T_1 \circ_{\mathcal{J}} T_2,$

(ii)   $\diamond_{\mathcal{J}} S_1 \subseteq \diamond_{\mathcal{J}} T_1.$

From inclusion monotonicity, plus the fact that $s \in S \Leftrightarrow [s, s] \subseteq S$, if $s \in S$ and $t \in T$, then for $\circ_{\mathcal{J}} \in \{+, \times\}$ and $\diamond_{\mathcal{J}} \in \{-, ^{-1}\}$, we obviously have $s \circ_{\mathbb{R}} t \in S \circ_{\mathcal{J}} T$ and $\diamond_{\mathbb{R}} s \in \diamond_{\mathcal{J}} S$.

At this point, let us introduce an abbreviation that we will make use of. Let $\boldsymbol{s} = (s_1, \ldots, s_i, \ldots, s_n)$ be an ordered real $n$-tuple, and let $\boldsymbol{S} = (S_1, \ldots, S_i, \ldots, S_n)$ and $\boldsymbol{T} = (T_1, \ldots, T_i, \ldots, T_n)$ be two ordered $n$-tuples of $\mathcal{J}$-numbers, then

$$\boldsymbol{S} \subseteq \boldsymbol{T} \Leftrightarrow (\forall i)(S_i \subseteq T_i),$$

$$\boldsymbol{s} \in \boldsymbol{S} \Leftrightarrow (\forall i)(s_i \in S_i).$$

In the following theorem, let $[s] \leq_{\mathcal{J}_{[s]}} [t] \Leftrightarrow s \leq_{\mathbb{R}} t$.

**Theorem 4.5 (Isomorphism Theorem for $\mathcal{J}$-Numbers ):** *The structure $\langle \mathcal{J}_{[s]}; +_{\mathcal{J}}, \times_{\mathcal{J}}; \leq_{\mathcal{I}_{[s]}} \rangle$ of point $\mathcal{J}$-numbers is isomorphic to the ordered field $\langle \mathbb{R}; +_{\mathbb{R}}, \times_{\mathbb{R}}; \leq_{\mathbb{R}} \rangle$ of real numbers.*

Two further results we will need are stated below (*Dawood, 2012*).

**Theorem 4.5 (Algebraic Operations for Point $\mathcal{J}$-Numbers):** *Let $S$ and $T$ be two $\mathcal{J}$-numbers. Then:*

(i) *The sum $S + T$ is a point $\mathcal{J}$-number iff each of $S$ and $T$ is a point $\mathcal{J}$-number, that is*
$$(\forall S, T \in \mathcal{J})\big(S + T \in \mathcal{J}_{[s]} \Leftrightarrow S \in \mathcal{J}_{[s]} \wedge T \in \mathcal{J}_{[s]}\big).$$

(ii) *The product $S \times T$ is a point $\mathcal{J}$-number iff each of $S$ and $T$ is a point $\mathcal{J}$-number, or at least one of $S$ and $T$ is $0_{\mathcal{J}}$, that is*
$$(\forall S, T \in \mathcal{J})\big(S \times T \in \mathcal{J}_{[s]} \Leftrightarrow \big(S \in \mathcal{J}_{[s]} \wedge T \in \mathcal{J}_{[s]}\big) \vee (S = 0_{\mathcal{J}} \vee T = 0_{\mathcal{J}})\big).$$

**Theorem 4.7 (Zero Divisors in $\mathcal{J}$-Numbers):** *Nonzero zero divisors do not exist in $\mathcal{J}$-arithmetic, that is*

$$(\forall S, T \in \mathcal{J})(S \times T = 0_{\mathcal{J}} \Rightarrow S = 0_{\mathcal{J}} \vee T = 0_{\mathcal{J}}).$$

Before turning to the axioms of the theory $\mathsf{T}_{\delta\mathcal{J}}$ of a differential $\mathcal{J}$-algebra, it is necessary for our purpose to formalize some analytic concepts within the framework of the theory $\mathsf{T}_{\mathcal{J}}$ of $\mathcal{J}$-numbers.

Before proceeding any further, let us agree on some basic notation. By an *n*-ary *real function* (in short, $\mathbb{R}$-function) we will always mean a function $q_{\mathbb{R}} : \mathcal{D}_{\mathbb{R}} \subseteq \mathbb{R}^n \mapsto \mathbb{R}$, and by an *n*-ary *interval function* (in short, $\mathcal{J}$-function) we will always mean a function $q_{\mathcal{J}} : \mathcal{D}_{\mathcal{J}} \subseteq \mathcal{J}^n \mapsto \mathcal{J}$. The $\mathbb{R}$-subscripted symbols $q_{\mathbb{R}}, u_{\mathbb{R}}, v_{\mathbb{R}}$ will designate $\mathbb{R}$-functions, while the $\mathcal{J}$-subscripted symbols $q_{\mathcal{J}}, u_{\mathcal{J}}, v_{\mathcal{J}}$ will designate $\mathcal{J}$-functions. For simplicity of notation, if the function type is apparent from the type of its variables(arguments), the subscripts " $\mathbb{R}$" and " $\mathcal{J}$" will usually be dropped. For instance, whenever unambiguous, we use the notations $q(s_1, \ldots, s_n)$ and $q(S_1, \ldots, S_n)$ for, respectively, an $\mathbb{R}$-function and a $\mathcal{J}$-function, which are both defined by the same rule. For $1 \leq i \leq n$ and $1 \leq j \leq k$, let $S_i$ and $A_j$ be respectively $\mathcal{J}$-variable symbols and $\mathcal{J}$-constant symbols. We denote by $q_{\mathcal{J}}\big(S_{i:n}; A_{j:k}\big)$ an *n*-ary (or multivariate) $\mathcal{J}$-function in the interval variables $S_i$ and the interval constants $A_j$. Similarly, we understand by $q_{\mathbb{R}}\big(s_{i:n}; a_{j:k}\big)$ an *n*-ary $\mathbb{R}$-function in the real variables $s_i$ and the real constants $a_j$. For instance,

$$q_{\mathcal{J}}\big(S_{i:2}; A_{j:2}\big) = q_{\mathcal{J}}(S_1, S_2; A_1, A_2) = A_1 S_1^2 + A_2 S_2,$$

is a binary $\mathcal{J}$-function whose variable arguments are $S_1$ and $S_2$, and whose constants are $A_1$ and $A_2$.

With a few exceptions, without loss of generality, the present discussion will be confined to unary functions only. For brevity, therefore, we will often adopt the standard notations $q(s)$ and $q(S)$ respectively for the unary functions $q\big(s; a_{j:k}\big)$ and $q\big(S; A_{j:k}\big)$. The sets of unary real and interval functions will be denoted by $\mathbb{R}_{\langle 1 \rangle}$ and $\mathcal{J}_{\langle 1 \rangle}$ respectively.

Next, we define the interval enclosure of a bounded set of real numbers.

**Definition 4.6 (Interval Enclosure of a Bounded Set):** *Let $\mathcal{A}$ be a bounded subset of $\mathbb{R}$. The interval enclosure of $\mathcal{A}$, denoted $\mathrm{E}_{\mathcal{J}}$, is defined to be*

$$\mathrm{E}_{\mathcal{J}}(\mathcal{A}) = \left[\inf(\mathcal{A}), \sup(\mathcal{A})\right].$$

Clearly, $\mathcal{A} \subseteq \mathrm{E}_{\mathcal{J}}(\mathcal{A})$. For instance $\mathrm{E}_{\mathcal{J}}(\{3,4,2\}) = [2,4]$ and $\mathrm{E}_{\mathcal{J}}([1,3[) = [1,3]$.

An important notion we will need is that of the image set of bounded subsets of $\mathbb{R}$, under an $n$-ary real-valued function. This notion is a special case of that of the corresponding $(n+1)$-ary relation on $\mathbb{R}$. More precisely, we have the following definition.

**Definition 4.7 (Image of Bounded Real Sets):** *Let $q$ be an $n$-ary function on $\mathbb{R}$, and for $(\mathbf{s}, t) \in q$, let $\mathbf{s} = (s_1, \ldots, s_n)$, with each $s_i$ is restricted to vary on a bounded set $\mathcal{S}_i \subset \mathbb{R}$, that is, $\mathbf{s}$ is restricted to vary on a set $\mathbf{S} \subset \mathbb{R}^n$. Then, the image of $\mathbf{S}$ (or the image of the sets $\mathcal{S}_k$) with respect to $q$, in symbols $\mathrm{I}_q$, is characterized to be*

$$\begin{aligned}
\mathcal{T} = \mathrm{I}_q(\mathbf{S}) &= \mathrm{I}_q(\mathcal{S}_1, \ldots, \mathcal{S}_n) \\
&= \{t \in \mathbb{R} \mid (\exists \mathbf{s} \in \mathbf{S})\,(\mathbf{s}qt)\} \\
&= \{t \in \mathbb{R} \mid (\exists_{i=1}^n s_i \in S_i)\,(t = q(s_1, \ldots, s_n))\} \subseteq \mathbb{R}.
\end{aligned}$$

*The preimage[7] $\mathbf{S}$ of $\mathcal{T}$ is characterized to be the image of $\mathcal{T}$ with respect to the converse $\widehat{q}$ of $q$. In other words*

$$\mathbf{S} = \mathrm{I}_{\widehat{q}}(\mathcal{T}) = \{\mathbf{s} \in \mathbb{R}^n \mid (\exists t \in \mathcal{T})\,(t\widehat{q}\mathbf{s})\}.$$

[7] From the fact that the converse relation $\widehat{q}$ is always definable, the preimage of a function $q$ is always definable, regardless of the definability of the inverse function $q^{-1}$.

Two notions essential for the investigation conducted in this article are those of a family of real functions and its image.

**Definition 4.8 (Real Family):** *For $1 \leq i \leq n$ and $1 \leq j \leq k$, an $n$-ary real family (a family of $n$-ary real functions, or in short, an $\mathbb{R}$-family), denoted $\mathbf{Q}_{\mathbb{R}}(s_{i:n}; a_{j:k})$, is a set of real functions $q_{\mathbb{R}}(s_{i:n}; a_{j:k})$ subject to the following conditions*

(i) *$q$ is a function rule,*

(ii) *$s_i$ are variable symbols varying on bounded subsets $\mathcal{S}_i$ of $\mathbb{R}$,*

(iii) *$a_j$ are constant symbols (coefficients) from bounded subsets $\mathcal{A}_j$ of $\mathbb{R}$, and*

(iv) *for each $a_j \in \mathcal{A}_j$, $q_{\mathbb{R}}(s_{i:n}; a_{j:k})$ is continuous on the sets $\mathcal{S}_i$. We understand by the converse of $\mathbf{Q}_{\mathbb{R}}$, denoted $\widehat{\mathbf{Q}}_{\mathbb{R}}$, the set of the converse relations $\widehat{q}$.*

Note that a real family is generated by one function rule, that is, the functions $q_{\mathbb{R}}(s_{i:n}; a_{j:k})$ in $\mathbf{Q}_{\mathbb{R}}$ all have the same rule $q$ but different constant arguments. If the sets $\mathcal{A}_j$ are singletons, then the family $\mathbf{Q}_{\mathbb{R}}$ reduces to exactly one $n$-ary real function. To clarify the matters, we give some examples.

**Example 4.1 (Real Families):** *The following are instances of real families.*

(i) *Let $\mathbf{Q}_{\mathbb{R}}$ be the family generated by the function rule $q_{\mathbb{R}}(s_{i:2}; a) = s_1^2 + as_2$, with the variables $s_1$ and $s_2$ vary respectively on the bounded sets $[2,4[$ and $[5,6]$ and the*

*constant $a$ is from the bounded set $\{3,7\}$. The family $\mathbf{Q}_{\mathbb{R}}$ has exactly the two binary functions*

$$q_{\mathbb{R}}(s_{i:2};3) = s_1^2 + 3s_2 \quad and \quad q_{\mathbb{R}}(s_{i:2};7) = s_1^2 + 7s_2.$$

(ii) *Let $\mathbf{U}_{\mathbb{R}}$ be the family generated by the function rule $u_{\mathbb{R}}(s;a) = as^4$, with the variable $s$ varies on the bounded set $[2,4[$ and the constant $a$ is from the bounded set $[1,2]$. The family $\mathbf{U}_{\mathbb{R}}$ has an infinite number of unary functions. Among these are, for example*

$$u_{\mathbb{R}}(s;1) = s^4, u_{\mathbb{R}}\left(s;\frac{3}{2}\right) = \frac{3s^4}{2}, ..., etc.$$

We characterize the image of a real family as follows.

**Definition 4.9 (Image of a Real Family):** *Let $\mathbf{Q}_{\mathbb{R}}$ be a real family generated by a function rule $t = q(s_{i:n}; a_{j:k})$, with $s_i \in \mathcal{S}_i$ and $a_j \in \mathcal{A}_j$. Then, the image of the family $\mathbf{Q}_{\mathbb{R}}$ (or the image of the sets $\mathcal{S}_i$ with respect to $\mathbf{Q}_{\mathbb{R}}$), denoted $\mathrm{I}_{\mathbf{Q}}$, is the union of the images of $\mathbf{S} = (\mathcal{S}_1, ..., \mathcal{S}_n) \subset \mathbb{R}^n$ with respect to each $q$ in $\mathbf{Q}_{\mathbb{R}}$ for all $a_j \in \mathcal{A}_j$. That is*

$$\mathcal{T} = \mathrm{I}_{\mathbf{Q}}(\mathbf{S}) = \mathrm{I}_{\mathbf{Q}}(\mathcal{S}_1, ..., \mathcal{S}_n)$$

$$= \{t \in \mathbb{R} | \left(\exists_{i=1}^n s_i \in \mathcal{S}_i\right)\left(\exists_{j=1}^k a_j \in \mathcal{A}_j\right)(t = q(s_{i:n}; a_{j:k}))\} \subseteq \mathbb{R}.$$

Obviously, for each $q$ in $\mathbf{Q}_{\mathbb{R}}$, $\mathrm{I}_q \subseteq \mathrm{I}_{\mathbf{Q}}$. An immediate consequence of definition 4.9 and the well-known extreme value theorem (see *Dawood, 2012*) is the following important property.

**Theorem 4.8 (Main Theorem of Image Evaluation):** *Let $\mathbf{Q}_{\mathbb{R}}$ be a real family generated by a function rule $q(s_{i:n}; a_{j:k})$, with $s_i \in \mathcal{S}_i$ and $a_j \in \mathcal{A}_j$. If $\mathcal{S}_i$ and $\mathcal{A}_j$ are real closed intervals, then the image $\mathrm{I}_{\mathbf{Q}}(\mathcal{S}_1, ..., \mathcal{S}_n)$ of $\mathcal{S}_i$, with respect to the family $\mathbf{Q}_{\mathbb{R}}$, is in turn a real closed interval such that*

$$\mathrm{I}_{\mathbf{Q}}(\mathcal{S}_1, ..., \mathcal{S}_n) = \left[ \min_{\substack{s_i \in \mathcal{S}_i \\ a_j \in \mathcal{A}_j}} q(s_{i:n}; a_{j:k}), \max_{\substack{s_i \in \mathcal{S}_i \\ a_j \in \mathcal{A}_j}} q(s_{i:n}; a_{j:k}) \right].$$

If the sets $\mathcal{A}_j$ of coefficients are singletons, then the family is in turn a *singleton* and the image of $\mathrm{I}_{\mathbf{Q}}$ reduces to the usual image $\mathrm{I}_q$ of a real function $q$ over real closed intervals

$$\mathrm{I}_q(\mathcal{S}_1, ..., \mathcal{S}_n) = \left[ \min_{s_i \in \mathcal{S}_i} q(s_1, ..., s_n), \max_{s_i \in \mathcal{S}_i} q(s_1, ..., s_n) \right].$$

By referring to definition 4.6, we can characterize the important notion of the interval extension of a real family.

**Definition 4.10 (Interval Extension of a Real Family):** *Let $\mathbf{Q}_{\mathbb{R}}$ be an $n$-ary real family generated by a function rule $q_{\mathbb{R}}(s_{i:n}; a_{j:k})$, with $s_i \in \mathcal{S}_i$ and $a_j \in \mathcal{A}_j$. We understand by an interval extension of $\mathbf{Q}_{\mathbb{R}}$ an $n$-ary interval function $q_{\mathcal{J}}(\mathcal{S}_{i:n}; \mathcal{A}_{j:k})$ of the same rule as $q_{\mathbb{R}}$, and whose arguments are $\mathcal{S}_i = \mathrm{E}_{\mathcal{J}}(\mathcal{S}_i)$ and $\mathcal{A}_j = \mathrm{E}_{\mathcal{J}}(\mathcal{A}_j)$.*

Clearly, if $\mathcal{S}_i$ and $\mathcal{A}_j$ are real closed intervals, then $S_i = \mathcal{S}_i$ and $A_j = \mathcal{A}_j$. We will henceforth deploy the predicate $\mathrm{Ext}(q_{\mathcal{J}}, \mathbf{Q}_{\mathbb{R}})$ to mean that an interval function $q_{\mathcal{J}}$ is the *interval extension* of the real family $\mathbf{Q}_{\mathbb{R}}$, or equivalently, the family $\mathbf{Q}_{\mathbb{R}}$ is the *real intension* of the interval function $q_{\mathcal{J}}$. If $\mathcal{A}_j$ are singletons, then the family $\mathbf{Q}_{\mathbb{R}}$ is a singleton and we call $q_{\mathcal{J}}$ a *simple extension* of $\mathbf{Q}_{\mathbb{R}}$. If $\mathcal{S}_i$ and $\mathcal{A}_j$ are singletons, then we call the point-valued interval function $q_{\mathcal{J}}$ a *point extension* of $\mathbf{Q}_{\mathbb{R}}$.

The following example will illustrate this point.

**Example 4.2 (Interval Extensions of Real Families):** *Recall the real families $\mathbf{Q}_{\mathbb{R}}$ and $\mathbf{U}_{\mathbb{R}}$ of example 4.1. The interval extensions of $\mathbf{Q}_{\mathbb{R}}$ and $\mathbf{U}_{\mathbb{R}}$ are given respectively by*

(i) $\quad q_{\mathcal{J}}(S_{i:2}; A) = S_1^2 + AS_2$, with $S_1 = \mathrm{E}_{\mathcal{J}}([2,4[) = [2,4]$, $S_2 = \mathrm{E}_{\mathcal{J}}([5,6]) = [5,6]$, and $A = \mathrm{E}_{\mathcal{J}}(\{3,7\}) = [3,7]$.

(ii) $\quad u_{\mathcal{J}}(S; A) = AS^4$, with $S = \mathrm{E}_{\mathcal{J}}([2,4[) = [2,4]$, and $A = \mathrm{E}_{\mathcal{J}}([1,2]) = [1,2]$.

The previous discussion faces us with the reasonable question: *does every interval function have a real intension?* In order to answer this, we next define what a *proper interval function* is.

**Definition 4.11 (Proper Interval Function):** *We say that an interval function $q_{\mathcal{J}} : \mathcal{D}_{\mathcal{J}} \subseteq \mathcal{J}^n \mapsto \mathcal{J}$ is proper, in symbols $\mathrm{Prop}(q_{\mathcal{J}})$, iff it is set-theoretically definable in terms of a real function of the same rule. That is*

$$\mathrm{Prop}(q_{\mathcal{J}}) \Leftrightarrow (\exists q_{\mathbb{R}})\left(q_{\mathcal{J}}(S_1, \ldots, S_n) = \{w \in \mathbb{R} \mid (\exists_{i=1}^n s_i \in S_i)(w = q_{\mathbb{R}}(s_1, \ldots, s_n))\}\right).$$

By definitions 4.11 and 4.5, the following result is derivable.

**Theorem 4.9 (Criteria for Proper Interval Functions):** *Let $\circ \in \{+, \times\}$ be a binary $\mathcal{J}$-operation and $\diamond \in \{-, ^{-1}\}$ be a definable unary $\mathcal{J}$-operation. Then, the following statements are true.*

(i) $\quad (\forall q_{\mathcal{J}}, u_{\mathcal{J}})(\mathrm{Prop}(q_{\mathcal{J}}) \wedge \mathrm{Prop}(u_{\mathcal{J}}) \Rightarrow \mathrm{Prop}(q_{\mathcal{J}} \circ u_{\mathcal{J}}))$,

(ii) $\quad (\forall q_{\mathcal{J}})(\mathrm{Prop}(q_{\mathcal{J}}) \Rightarrow \mathrm{Prop}(\diamond q_{\mathcal{J}}))$,

(iii) $\quad (\forall q_{\mathcal{J}}, u_{\mathcal{J}})(\mathrm{Prop}(q_{\mathcal{J}}) \wedge \mathrm{Prop}(u_{\mathcal{J}}) \Rightarrow \mathrm{Prop}(q_{\mathcal{J}}(u_{\mathcal{J}})))$.

In accordance with definition 4.11 and its previous consequence, we have then the following important result.

**Theorem 4.10 (Intensionality of an Interval Function):** *An interval function is intensionable iff it is proper. In other words*

$$\mathrm{Prop}(q_{\mathcal{J}}) \Leftrightarrow (\exists \mathbf{Q}_{\mathbb{R}})(\mathrm{Ext}(q_{\mathcal{J}}, \mathbf{Q}_{\mathbb{R}})).$$

For example all *elementary* interval functions are intensionable. On the contrary, *degenerate* functions such as the *midpoint* or *radius* of an interval are not proper and accordingly not intensionable. Definition 4.11 and the deductions from it can be easily generalized to proper $\mathcal{J}^m$-valued functions, in which case their intensions will be families of $\mathbb{R}^m$-valued functions.

Toward axiomatizing a theory of a differential interval algebra, it remains to formalize the notions of differentiability of a real family and of an interval function. Henceforth, we will

consider only families of unary real functions and their interval extensions. Accordingly, when there is no potential for ambiguity, we will write $\mathbf{Q}(s)$, or simply $\mathbf{Q}$, for the unary real family $\mathbf{Q}(s; a_{j:k})$.

Next, we extend the differential operator to families of unary real functions.

**Definition 4.12 (Differential Operator for a Real Family):** *Let* $\mathbf{Q}_{\mathbb{R}}(s; a_{j:k})$ *be a unary real family in the real variable* $s$ *and the real constants* $a_j$. *For a nonnegative integer* $n$, *the n-differential operator of* $\mathbf{Q}_{\mathbb{R}}(s; a_{j:k})$, *denoted* $\delta^n \mathbf{Q}_{\mathbb{R}}(s; a_{j:k})$, *is defined to be the set of all real functions* $\delta^n q(s; a_{j:k})$ *for every* $q \in \mathbf{Q}_{\mathbb{R}}$ *and every constant* $a_j$.

We have yet nothing to tell us if a real family is differentiable. The following two definitions introduce, respectively, the notions of differentiability and continuous differentiability of a unary real family $\mathbf{Q}(s; a_{j:k})$.

**Definition 4.13 (Differentiability of a Real Family):** *A unary real family* $\mathbf{Q}_{\mathbb{R}}(s; a_{j:k})$ *is n-differentiable at a real constant* $s_0$, *in symbols* $\mathrm{diff}^n(\mathbf{Q}, s_0)$, *iff for every* $q$ *in* $\mathbf{Q}_{\mathbb{R}}$, $s_0 \in \mathrm{dom}(q)$, *and* $q$ *is n-differentiable at* $s_0$. *That is*

$$\mathrm{diff}^n(\mathbf{Q}, s_0) \Leftrightarrow (\forall q \in \mathbf{Q})(s_0 \in \mathrm{dom}(q) \wedge \mathrm{diff}^n(q, s_0)).$$

**Definition 4.14 (Continuous Differentiability of a Real Family):** *A unary real family* $\mathbf{Q}_{\mathbb{R}}(s; a_{j:k})$ *is continuously n-differentiable at a real constant* $s_0$, *in symbols* $\mathrm{cdiff}^n(\mathbf{Q}, s_0)$, *iff for every* $q$ *in* $\mathbf{Q}_{\mathbb{R}}$, $s_0 \in \mathrm{dom}(q)$, *and* $q$ *is continuously n-differentiable at* $s_0$. *That is*

$$\mathrm{cdiff}^n(\mathbf{Q}, s_0) \Leftrightarrow (\forall q \in \mathbf{Q})(s_0 \in \mathrm{dom}(q) \wedge \mathrm{cdiff}^n(q, s_0)).$$

In accordance with the above concepts, the differential operator for interval functions is then definable.

**Definition 4.15 (Interval Differential Operator):** *Let* $n \geq 0$, *and let* $q(S)$ *be a unary interval function that has a real intension the family* $\mathbf{Q}(s)$. *The n-differential* $\mathcal{J}$-*operator of* $q(S)$, *denoted* $\delta^n q(S)$, *is characterized to be the interval extension of* $\delta^n \mathbf{Q}(s)$. *In other words, let* $\delta^n \mathbf{Q}(s) = \mathbf{U}(s)$, *then* $\delta^n q(S) = u(S)$.

In a manner analogous to differentiability in $\mathbb{R}$, the interval differentiability predicate is definable as follows.

**Definition 4.16 (Interval Differentiability Predicate):** *Let* $n \geq 0$, *let* $q \in \mathcal{J}_{\langle 1 \rangle}$, *and let* $S_0 \in \mathrm{dom}(q)$ *be a* $\mathcal{J}$-*constant symbol. The ternary n-differentiability* $\mathcal{J}$-*predicate, denoted* $\mathrm{diff}^n(q, S_0)$, *is defined by*

$$\mathrm{diff}^n(q, S_0) \Leftrightarrow \delta^n q(S_0) \in \mathcal{J}.$$

*If* $\mathrm{diff}^n(q, S_0)$ *is* true, *then the interval function* $q$ *is said to be n-differentiable at the closed interval* $S_0$.

Throughout this article, we will employ the following abbreviation.

$$\mathrm{diff}^n(q_1, q_2, \dots, q_k)_{S_0} \Leftrightarrow \mathrm{diff}^n(q_1, S_0) \wedge \mathrm{diff}^n(q_2, S_0) \wedge \dots \wedge \mathrm{diff}^n(q_k, S_0).$$

By means of definitions 4.15 and 4.16 plus a simple continuity argument, we have the following theorem that establishes the criteria for *interval differentiability*.

**Theorem 4.11 (Interval Differentiability Criteria):** *An interval function $q_{\mathcal{J}} \in \mathcal{J}_{\langle 1 \rangle}$ is n-differentiable at a $\mathcal{J}$-number $S_0$ if and only if*

(i) *$q_{\mathcal{J}}(S)$ is proper with a real intension $\mathbf{Q}(s)$, and*

(ii) *$\mathbf{Q}(s)$ is continuously n-differentiable at every $s_0 \in S_0$.*

From the fact that images of $\mathbb{R}$-functions are inclusion isotonic (*Dawood, 2012*), we have the next key result concerning interval enclosures of $\mathbb{R}$-families.

**Theorem 4.12 (Image Enclosure of a Real Family):** *Let $q(s)$ be a real function in a family $\mathbf{Q}(s)$, with $s$ is restricted to vary on a real closed interval $S_0$, and let $q(S_0)$ be the interval extension of $\mathbf{Q}(s)$ at $S_0$. The following two sentences are true.*

(i) $\left(\forall q \in \mathbf{Q}\right)\left(\mathrm{I}_q(S_0) \subseteq \mathrm{I}_{\mathbf{Q}}(S_0) \subseteq q(S_0)\right),$

(ii) $\left(\forall \delta q \in \delta\mathbf{Q}\right)\left(\mathrm{I}_{\delta q}(S_0) \subseteq \mathrm{I}_{\delta\mathbf{Q}}(S_0) \subseteq \delta q(S_0)\right).$

Moreover, *finer* enclosures of real families can be obtained *via* the subdivision method. The following corollary is implied by theorem 4.12.

**Corollary 4.1 (Subdivision Enclosure of a Real Family):** *Recall the notation used in theorem 4.12, and let $S_0$ be subdivided into $n \geq 1$ subintervals. Then*

$$\mathrm{I}_{\mathbf{Q}}(S_0) \subseteq \cup_{i=1}^{n} q(S_i) \subseteq q(S_0).$$

Obviously, $\mathrm{I}_{\mathbf{Q}}(S_0) = \lim_{n\to\infty} \cup_{i=1}^{n} q(S_i)$.

To the best of our knowledge, in all interval literature, an interval-valued function is assumed to have *singleton* (real) constants and accordingly an interval function might be only an extension of a *single* real function. An interesting and important observation from the above discussion is that this presumption introduces an *unnecessary restriction* to the semantic of an interval function in the general sense. As above characterized, a proper interval function $q_{\mathcal{J}}\left(S_{i:n}; A_{j:k}\right)$ is an extension of a *whole* family of real functions and this family is a singleton if, and only if, the interval constants $A_j$ are singletons.

With the aid of the notions now at hand, we can then axiomatize the theory $\mathsf{T}_{\delta\mathcal{J}}$ of a differential interval algebra (henceforth a *differential $\mathcal{J}$-algebra*).

**Definition 4.17 (Theory of a Differential Interval Algebra):** *Let $\sigma = \{+, \times; -, ^{-1}; 0, 1\}$ be a class of non-logical signs, and let $\mathsf{T}_{\mathcal{J}}$ be the theory of an interval $\mathsf{S}$-semiring $\mathfrak{J} = \langle \mathcal{J}; \sigma^{\mathfrak{J}} \rangle$. The theory $\mathsf{T}_{\delta\mathcal{J}}$ of a differential $\mathcal{J}$-algebra $\mathfrak{J}_{\delta} = \langle \mathcal{J}; \sigma^{\mathfrak{J}}; \delta \rangle$ is the deductive closure of $\mathsf{T}_{\mathcal{J}}$ together with the following two axioms.*

(i) $\left(\forall q, u \in \mathcal{J}_{\langle 1 \rangle}\right)\left(\delta\left(q + u\right) = \delta q + \delta u\right),$

(ii) $\left(\forall q, u \in \mathcal{J}_{\langle 1 \rangle}\right)\left(\delta\left(q \times u\right) = q \times \delta u + u \times \delta q\right).$

Consider the constant interval functions $q(S) = 0_{\mathcal{J}}$ and $u(S) = 1_{\mathcal{J}}$. With the aid of definition 4.15, obviously $\delta(0_{\mathcal{J}}) = \delta(1_{\mathcal{J}}) = 0_{\mathcal{J}}$. More generally, for any interval constant symbol $A$, $\delta(A) = 0_{\mathcal{J}}$ and $\delta(AS) = A$. Accordingly, the set $\mathcal{J}$ can be defined thus: $\mathcal{J} = \{q \in \mathcal{J}_{\langle 1 \rangle} | \delta q = 0_{\mathcal{J}}\}$. On grounds of definition 4.15 and axioms (i) and (ii) of the preceding definition, further properties of interval differentiation can be derived analogously.

# A CATEGORICAL AXIOMATIZATION OF INTERVAL DIFFERENTIATION ARITHMETIC

Building on the system $\mathsf{T}_{\delta\mathcal{J}}$ of a differential $\mathcal{J}$-algebra axiomatized in the previous section, the present section provides a rigorous mathematical foundation for *interval differentiation arithmetic* (henceforth $\Delta_{\mathcal{J}}$-*arithmetic*). We are almost ready to lay out an axiom system for the theory $\mathsf{T}_{\Delta_{\mathcal{J}}}$ of *interval differentiation numbers* (henceforth $\Delta_{\mathcal{J}}$-*numbers*) as a two-sorted extension of $\mathsf{T}_{\delta\mathcal{J}}$. By virtue of the mathematical underpinnings presented in 'A differential interval algebra', we axiomatize, in the present section, the basic operations of $\mathsf{T}_{\Delta_{\mathcal{J}}}$ and prove some of their fundamental properties. Moreover, we prove categoricity and consistency of $\Delta_{\mathcal{J}}$-arithmetic.

An obvious starting point is to define *interval differentiation n-tuples*.

**Definition 5.1 (Interval Differentiation $n$-Tuples):** *Let* $\mathfrak{J}_\delta = \langle \mathcal{J}; \sigma^{\mathfrak{J}}; \delta \rangle$ *be a differential* $\mathcal{J}$*-algebra, let* $q$ *be a unary* $\mathcal{J}$*-function, and for an integer* $n \geq 0$*, let* $\mathcal{J}^n$ *be the* $n$*-th Cartesian power of* $\mathcal{J}$*. The set of all interval differentiation* $n$*-tuples over* $\mathcal{J}$*, with respect to an individual* $\mathcal{J}$*-constant* $S_0 \in \mathcal{J}$*, is characterized to be*

$$^n\mathbf{U}_{\mathcal{J}} = \left\{ \mathsf{Q} \in \mathcal{J}^{n+1} \,|\, (\exists q \in \mathcal{J}_{\langle 1 \rangle}) \left( \begin{array}{c} \mathsf{Q} = \left( \delta^0 q(S_0), \delta^1 q(S_0), \ldots, \delta^n q(S_0) \right) \\ \wedge\, S_0 \in \mathrm{dom}(q) \wedge \mathrm{diff}^n(q, S_0) \end{array} \right) \right\}.$$

An interval differentiation $n$-tuple is thus an ordered $n$-tuple of $\mathcal{J}$-constants. Hereafter, we will usually write $q, \delta q, \ldots, \delta^n q$ for $\delta^0 q(S_0), \delta^1 q(S_0), \ldots, \delta^n q(S_0)$, respectively. The present article is concerned with *dyadic* interval differentiation tuples, that is $n$-tuples with $n = 1$; and we will hereon adopt the name "*interval differentiation numbers*" (" $\Delta_{\mathcal{J}}$-*numbers*", or " $\Delta_{\mathcal{J}}$-*pairs* ") for dyadic interval differentiation tuples. Let $\mathbf{U}_{\mathcal{J}}$ designate the set of $\Delta_{\mathcal{J}}$-numbers at some $\mathcal{J}$-constant $S_0$, and let the letters $\mathsf{Q}$, $\mathsf{U}$, and $\mathsf{V}$, or equivalently the pairs $(q, \delta q)_{S_0}$, $(u, \delta u)_{S_0}$, and $(v, \delta v)_{S_0}$, be variable symbols varying on the set $\mathbf{U}_{\mathcal{J}}$ of $\Delta_{\mathcal{J}}$-pairs. Also, let the letters $\mathsf{A}$, $\mathsf{B}$, and $\mathsf{C}$, or equivalently $(a, 0_{\mathcal{J}})_{S_0}$, $(b, 0_{\mathcal{J}})_{S_0}$, and $(c, 0_{\mathcal{J}})_{S_0}$, designate constants of $\mathbf{U}_{\mathcal{J}}$. In particular, we use $1_{\mathbf{U}_{\mathcal{J}}}$ to designate the $\Delta_{\mathcal{J}}$-number $(1_{\mathcal{J}}, 0_{\mathcal{J}})_{S_0}$ and $0_{\mathbf{U}_{\mathcal{J}}}$ to designate the $\Delta_{\mathcal{J}}$-number $(0_{\mathcal{J}}, 0_{\mathcal{J}})_{S_0}$. Moreover, it is convenient for our purpose to define a proper subset of $\mathbf{U}_{\mathcal{J}}$ as

$$\mathbf{U}_{(\mathcal{J},0)} = \left\{ \mathsf{Q} \in \mathbf{U}_{\mathcal{J}} \,|\, \mathsf{Q} = (q, 0_{\mathcal{J}})_{S_0} \right\}.$$

We are now ready to axiomatize the theory $\mathsf{T}_{\Delta_{\mathcal{J}}}$ of an *interval differentiation algebra* (or a $\Delta_{\mathcal{J}}$-*algebra*) over an interval $\mathsf{S}$-*semiring*.

**Definition 5.2 (Theory of Interval Differentiation Algebra):** *Let* $\sigma = \{+, \times; -, ^{-1}; 0, 1\}$ *be a class of non-logical signs, and let* $(q, \delta q)_{S_0}$*,* $(u, \delta u)_{S_0}$*, and* $(v, \delta v)_{S_0}$ *be in* $\mathbf{U}_{\mathcal{J}}$*. An interval differentiation algebra (or, in short, a* $\Delta_{\mathcal{J}}$*-algebra) over a differential* $\mathcal{J}$*-algebra* $\mathfrak{J}_d = \langle \mathcal{J}; \sigma^{\mathfrak{J}}; \delta \rangle$ *is a two-sorted structure* $\mathfrak{U}_{\mathcal{J}} = \langle \mathbf{U}_{\mathcal{J}}; \mathcal{J}; \sigma^{\mathfrak{U}_{\mathcal{J}}} \rangle$*. The theory* $\mathsf{T}_{\Delta_{\mathcal{J}}}$ *of* $\mathfrak{U}_{\mathcal{J}}$ *is the deductive closure of the system* $\mathsf{T}_{\delta\mathcal{J}}$ *of* $\mathfrak{J}_d$ *and the following set of axioms.*

(IDA1)  *$\Delta_{\mathcal{J}}$-equality.* $(q, \delta q)_{S_0} =_{\mathbf{U}_{\mathcal{J}}} (u, \delta u)_{S_0} \Leftrightarrow q(S_0) =_{\mathcal{J}} u(S_0) \wedge \delta q(S_0) =_{\mathcal{J}} \delta u(S_0)$,

(IDA2)  *Binary $\Delta_{\mathcal{J}}$-operations.* $\circ \in \{+, \times\} \Rightarrow (q, \delta q)_{S_0} \circ_{\mathbf{U}_{\mathcal{J}}} (u, \delta u)_{S_0} = \left( q \circ_{\mathcal{J}} u, \delta(q \circ_{\mathcal{J}} u) \right)_{S_0}$,

(IDA3)  *Unary $\Delta_{\mathcal{J}}$-operations.* $\diamond \in \{-\} \vee \left( \diamond \in \{^{-1}\} \wedge 0_{\mathcal{J}} \nsubseteq q(S_0) \right) \Rightarrow \diamond_{\mathbf{U}_{\mathcal{J}}} (q, \delta q)_{S_0} = \left( \diamond_{\mathcal{J}} q, \delta(\diamond_{\mathcal{J}} q) \right)_{S_0}$.

The intended model of the theory $T_{\Delta_{\mathcal{J}}}$ corresponds the sets " $\mathcal{J}$" and " $U_{\mathcal{J}}$" to the sets of $\mathcal{J}$-*numbers* and $\Delta_{\mathcal{J}}$-*numbers*, respectively, and the symbols " $\circ_{\mathcal{J}}$", and " $\diamond_{\mathcal{J}}$" to the binary and unary $\mathcal{J}$-operations. When the context is clear, for simplicity henceforth, we will drop the subscripts " $U_{\mathcal{J}}$", " $\mathcal{J}$", and " $S_0$". Also, we will usually write the algebraic structure $\mathfrak{U}_{\mathcal{J}}$ as $\langle U_{\mathcal{J}}; +_{U_{\mathcal{J}}}, \times_{U_{\mathcal{J}}}; 0_{U_{\mathcal{J}}}, 1_{U_{\mathcal{J}}} \rangle$, omitting the set $\mathcal{J}$.

The *inclusion* and *membership* relations for $\Delta_{\mathcal{J}}$-numbers can be defined as follows.

**Definition 5.3 (Inclusion Relation on $\Delta_{\mathcal{J}}$-Numbers):** *The inclusion relation on $\Delta_{\mathcal{J}}$-numbers, denoted $\subseteq_{U_{\mathcal{J}}}$, is defined as follows.*

$$\left( \forall Q, U \in U_{\mathcal{J}} \right) \left( Q \subseteq_{U_{\mathcal{J}}} U \Leftrightarrow q_{\mathcal{J}}(S_0) \subseteq u_{\mathcal{J}}(S_0) \wedge \delta q_{\mathcal{J}}(S_0) \subseteq \delta u_{\mathcal{J}}(S_0) \right).$$

**Definition 5.4 (Membership Relation in $\Delta_{\mathcal{J}}$-Numbers):** *The membership relation in $\Delta_{\mathcal{J}}$-numbers, denoted $\in_{U_{\mathcal{J}}}$, is defined as follows.*

$$\left( \forall q \in U_{\mathbb{R}} \right) \left( \forall Q \in U_{\mathcal{J}} \right) \left( q \in_{U_{\mathcal{J}}} Q \Leftrightarrow q_{\mathbb{R}}(s_0) \in q_{\mathcal{J}}(S_0) \wedge \delta q_{\mathbb{R}}(s_0) \in \delta q_{\mathcal{J}}(S_0) \right).$$

An important notion for our purposes is that of a *point $\Delta_{\mathcal{J}}$-number*.

**Definition 5.5 (Point $\Delta_{\mathcal{J}}$-Number):** *By a point (or singleton) $\Delta_{\mathcal{J}}$-number, denoted $[q] = \left( [q], [\delta q] \right)_{S_0}$, we understand a $\Delta_{\mathcal{J}}$-number whose all components are point intervals, that is $[q](S_0)$ and $[\delta q](S_0)$ are in $\mathcal{J}_{[s]}$.*

The set of all point $\Delta_{\mathcal{J}}$-numbers will be denoted by $U_{[q]}$. In the sequel, we will make use of the following theorem that establishes the criteria when a $\Delta_{\mathcal{J}}$-number is a singleton.

**Theorem 5.1 (Criteria for Point $\Delta_{\mathcal{J}}$-Numbers):** *A $\Delta_{\mathcal{J}}$-number $(q, \delta q)_{S_0}$ is point iff*

(i)  *$q$ is a constant point-valued function, that is $q = [c] \in \mathcal{J}_{[s]}$, or*

(ii)  *$S_0 = [s_0] \in \mathcal{J}_{[s]}$ and each constant in the rule of $q$ is a point interval.*

**Proof** The proof is immediate from theorem 4.6. $\square$

By means of definitions 4.15 plus the rules of differential sum and product, axiomatized in definitions 4.17, the following theorem is easily derivable from the theory $T_{\Delta_{\mathcal{J}}}$.

**Theorem 5.2 (Algebraic Operations of $\Delta_{\mathcal{J}}$-Numbers):** *Let $(q, \delta q)_{S_0}$ and $(u, \delta u)_{S_0}$ be two $\Delta_{\mathcal{J}}$-numbers. Then, the binary and unary $\Delta_{\mathcal{J}}$-operations are formulated as follows.*

(i)  *$\Delta_{\mathcal{J}}$-addition. $(q, \delta q) +_{U_{\mathcal{J}}} (u, \delta u) = (q +_{\mathcal{J}} u, \delta q +_{\mathcal{J}} \delta u)$,*

(ii)  *$\Delta_{\mathcal{J}}$-multiplication. $(q, \delta q) \times_{U_{\mathcal{J}}} (u, \delta u) = (q \times_{\mathcal{J}} u, \delta q \times_{\mathcal{J}} u +_{\mathcal{J}} q \times_{\mathcal{J}} \delta u)$,*

(iii)  *$\Delta_{\mathcal{J}}$-negation. $-_{U_{\mathcal{J}}} (q, \delta q) = (-_{\mathcal{J}} q, -_{\mathcal{J}} \delta q).$,*

(iv)  *$\Delta_{\mathcal{J}}$-reciprocal. $0_{\mathcal{J}} \not\subseteq q(S_0) \Rightarrow (q, \delta q)^{-1_{U_{\mathcal{J}}}} = (q^{-1_{\mathcal{J}}}, -_{\mathcal{J}} q^{-2} \times_{\mathcal{J}} \delta q)$.*

To complete our characterization of $\Delta_{\mathcal{J}}$-arithmetic, we define as customary $\Delta_{\mathcal{J}}$-subtraction and $\Delta_{\mathcal{J}}$-division respectively as $Q - U = Q + (-U)$ and $Q \div U = Q \times \left( U^{-1} \right)$.

With the aid of the meta-theoretic notions characterized in definitions 2.2–2.4, we are able to proceed towards proving three important meta-theorems about the theory $T_{\Delta_{\mathcal{J}}}$ of $\Delta_{\mathcal{J}}$-numbers, concerning respectively *existence*, *categoricity* and *consistency* of a $\Delta_{\mathcal{J}}$-algebra.

**Theorem 5.3 (*Existence of a $\Delta_{\mathcal{J}}$-Algebra*):** There exists at least one $\Delta_{\mathcal{J}}$-algebra.

**Proof** Since the theory $\mathsf{T}_{\mathcal{J}}$ of a $\mathcal{J}$-algebra has the model $\langle\mathcal{J}_{\mathbb{R}};\sigma^{\mathfrak{J}}\rangle$ of $\mathcal{J}$-numbers, it follows that the theory $\mathsf{T}_{\Delta_{\mathcal{J}}}$ has a model $\langle\mathbf{U}_{\mathcal{J}};\mathcal{J};\sigma^{\mathfrak{U}_{\mathcal{J}}}\rangle$, and thus existence of a $\Delta_{\mathcal{J}}$-algebra is proved. □

**Theorem 5.4 (Categoricity of $\Delta_{\mathcal{J}}$-Arithmetic):** The theory $\mathsf{T}_{\Delta_{\mathcal{J}}}$ of $\Delta_{\mathcal{J}}$-numbers is categorical.

**Proof** The theorem follows from the categoricity of the theory $\mathsf{T}_{\mathcal{J}}$ of interval algebra by an argument analogous to the one used in theorem 4.1. □

That is, the theory $\mathsf{T}_{\Delta_{\mathcal{J}}}$ *uniquely* characterizes the algebra of $\Delta_{\mathcal{J}}$-numbers, and the structure $\langle\mathbf{U}_{\mathcal{J}};+_{\mathbf{U}_{\mathcal{J}}},\times_{\mathbf{U}_{\mathcal{J}}};0_{\mathbf{U}_{\mathcal{J}}},1_{\mathbf{U}_{\mathcal{J}}}\rangle$ is, up to isomorphism, the only possible model of $\mathsf{T}_{\Delta_{\mathcal{J}}}$. To reiterate, in accord to theorem 5.4, the system $\mathsf{T}_{\Delta_{\mathcal{J}}}$, axiomatized in definition 5.2, is the "best" axiomatization of $\Delta_{\mathcal{J}}$-numbers, in the sense that it rightly accounts, up to isomorphism, for *every structure* of $\Delta_{\mathcal{J}}$-arithmetic.[8]

The next theorem establishes the consistency of the theory $\mathsf{T}_{\Delta_{\mathcal{J}}}$ of $\Delta_{\mathcal{J}}$-numbers.

**Theorem 5.5 (Consistency of $\Delta_{\mathcal{J}}$-Arithmetic):** The theory $\mathsf{T}_{\Delta_{\mathcal{J}}}$ of $\Delta_{\mathcal{J}}$-numbers is consistent.

**Proof** In accord to definition 2.4, the proof is immediate from theorem 5.3. The theory $\mathsf{T}_{\Delta_{\mathcal{J}}}$ is satisfiable by the model $\langle\mathbf{U}_{\mathcal{J}};+_{\mathbf{U}_{\mathcal{J}}},\times_{\mathbf{U}_{\mathcal{J}}};0_{\mathbf{U}_{\mathcal{J}}},1_{\mathbf{U}_{\mathcal{J}}}\rangle$ and thus is consistent. □

Owing to the categoricity theorem for $\mathsf{T}_{\Delta_{\mathcal{J}}}$, the algebraic properties of $\mathcal{J}$-numbers are naturally assumed priori. Therefore, whenever unambiguous, hereon we will use these properties without further mention.

Noteworthy, by virtue of the theory developed so far, we have the profound results that each $\Delta_{\mathcal{J}}$-number represents a *guaranteed interval enclosure* of the image of a *whole family* of $\mathbb{R}$-functions and their derivatives and accordingly that a $\Delta_{\mathcal{J}}$-number is an interval extension of every $\Delta_{\mathbb{R}}$-number that corresponds to each function in the real family (See 'Machine implementation of interval auto-differentiation' for clarifying numerical examples). In consequence of theorem 4.12, these important results are made precise in the following immediate theorem and its corollary.

**Theorem 5.6 (Differential Enclosure of a Real Family):** *Let* $\mathbf{Q}$ *be a unary real family continuously differentiable on a real closed interval* $S_0$ *and let* $q_{\mathcal{J}}$ *be its interval extension. Then, for every* $q_{\mathbb{R}}$ *in* $\mathbf{Q}$

$$\left(\mathrm{I}_{q_{\mathbb{R}}}(S_0),\mathrm{I}_{\delta q_{\mathbb{R}}}(S_0)\right)\subseteq_{\mathbf{U}_{\mathcal{J}}}\left(\mathrm{I}_{\mathbf{Q}}(S_0),\mathrm{I}_{\delta\mathbf{Q}}(S_0)\right)\subseteq_{\mathbf{U}_{\mathcal{J}}}\left(q_{\mathcal{J}}(S_0),\delta q_{\mathcal{J}}(S_0)\right).$$

**Corollary 5.1 (Interval Extension of a $\Delta_{\mathbb{R}}$-Number):** *Let* $q$ *be a real function continuously differentiable on a real closed interval* $S_0$. *Then, for every* $s_0 \in S_0$,

$$\left(q_{\mathbb{R}},\delta q_{\mathbb{R}}\right)_{s_0}\in_{\mathbf{U}_{\mathcal{J}}}\left(q_{\mathcal{J}},\delta q_{\mathcal{J}}\right)_{S_0}.$$

[8]Categoricity is a bedrock of mathematics. For further details on the key role of categoricity in logic and mathematics, see, *e.g.*, *Corcoran, 1980*, *Dawood & Megahed, 2019*, and *Shapiro, 1985*.

Finally, let us note that we can get *sharper* enclosures of the pair $\left(I_{\mathbf{Q}}(S_0), I_{\delta\mathbf{Q}}(S_0)\right)$ with the aid of the *subdivision method*. In consequence of theorems 5.2 and 5.6 we are led to the following theorem.

**Theorem 5.7 (Subdivision Theorem for $\Delta_{\mathcal{J}}$-Numbers):** *Recall the notation used in theorem 5.6, and let $S_0$ be subdivided into $n \geq 1$ subintervals. Then*

$$\left(I_{\mathbf{Q}}(S_0), I_{\delta\mathbf{Q}}(S_0)\right) \subseteq_{\mathbf{U}_{\mathcal{J}}} \bigcup_{i=1}^{n} \left(q_{\mathcal{J}}(S_i), \delta q_{\mathcal{J}}(S_i)\right) \subseteq_{\mathbf{U}_{\mathcal{J}}} \left(q_{\mathcal{J}}(S_0), \delta q_{\mathcal{J}}(S_0)\right).$$

*Moreover,* $\left(I_{\mathbf{Q}}(S_0), I_{\delta\mathbf{Q}}(S_0)\right) = \lim_{n \to \infty} \bigcup_{i=1}^{n} \left(q_{\mathcal{J}}(S_i), \delta q_{\mathcal{J}}(S_i)\right).$

## DIFFERENTIATION EXTENSION OF INTERVAL FUNCTIONS AND HIGHER-ORDER AUTO-DIFFERENTIABILITY

We aim to fully address and compute higher order and partial auto-derivatives using only *dyadic* $\Delta_{\mathcal{J}}$-numbers ($\Delta_{\mathcal{J}}$-pairs), and without resorting to defining any sort of $n$-dimensional *Grassmann algebras*. Towards this end, we are to extend the theory $\mathsf{T}_{\Delta_{\mathcal{J}}}$, by introducing the notion of a *differentiation extension* of $\mathcal{J}$-functions, characterizing *differentiability* for $\Delta_{\mathcal{J}}$-numbers, and establishing their *differentiability conditions*.

In view of our definition of $\Delta_{\mathcal{J}}$-numbers, the following alternate characterization of interval differentiability is at our disposal.

$$\text{diff}^1\left(q, S_0\right) \Leftrightarrow \delta q(S_0) \in \mathcal{J}_{\mathbb{R}} \Leftrightarrow \left(q, \delta q\right)_{S_0} \in \mathbf{U}_{\mathcal{J}}.$$

In order to have $\Delta_{\mathcal{J}}$-functions beyond the *rational functions* defined in 'A categorical axiomatization of interval differentiation arithmetic', an *extension principle* should be introduced. Thus we require to extend $\mathcal{J}$-functions to $\Delta_{\mathcal{J}}$-functions. In accord to the above characterization, we have the next definition.

**Definition 6.1 (Differentiation Extensions of $\mathcal{J}$-Functions):** *For $k \in \{1, \ldots, n\}$, let $u_k \in \mathcal{J}_{\langle 1 \rangle}$ be differentiable at $S_0 \in \text{dom}(u_k)$, that is for each $u_k$ there is $\mathsf{U}_k = (u_k(S_0), \delta u_k(S_0)) \in \mathbf{U}_{\mathcal{J}}$. Let $\mathcal{Q}_{\mathcal{J}}(u_1, \ldots, u_n)$ be an $n$-place $\mathcal{J}$-function of $u_1, \ldots, u_n$ which is differentiable at $S_0$. A differentiation extension of $\mathcal{Q}_{\mathcal{J}}$ is an $n$-place $\Delta_{\mathcal{J}}$-function $\mathcal{Q}_{\mathbf{U}_{\mathcal{J}}}(\mathsf{U}_1, \ldots, \mathsf{U}_n)$ defined to be*

$$\mathcal{Q}_{\mathbf{U}_{\mathcal{J}}}(\mathsf{U}_1, \ldots, \mathsf{U}_n) = (\mathcal{Q}_{\mathcal{J}}(u_1, \ldots, u_n), \delta\mathcal{Q}_{\mathcal{J}}(u_1, \ldots, u_n)),$$

*and obtained from $\mathcal{Q}_{\mathcal{J}}$ by replacing, in $\mathcal{Q}_{\mathcal{J}}$, each occurrence of a $\mathcal{J}$-function symbol $u_k$ by the corresponding $\Delta_{\mathcal{J}}$-variable symbol $\mathsf{U}_k$.*

The definition is so framed that since $\text{diff}^1(\mathcal{Q}_{\mathcal{J}}, S_0)$ is true, the differentiation extension $\mathcal{Q}_{\mathbf{U}_{\mathcal{J}}}$ of $\mathcal{Q}_{\mathcal{J}}$ is in $\mathbf{U}_{\mathcal{J}}$. Thus, $\mathcal{Q}_{\mathcal{J}}$ and $\mathcal{Q}_{\mathbf{U}_{\mathcal{J}}}$ are both defined by the same symbolic rule but with different types of arguments (variables); $\mathcal{Q}_{\mathcal{J}}$ is a $\mathcal{J}$-function whereas $\mathcal{Q}_{\mathbf{U}_{\mathcal{J}}}$ is a $\Delta_{\mathcal{J}}$-function. By analogy with rational $\mathcal{J}$-functions, a *rational $\Delta_{\mathcal{J}}$-function* is a (multivariate) $\Delta_{\mathcal{J}}$-function obtained by the application of a finite number of the binary and unary algebraic $\Delta_{\mathcal{J}}$-operations $\circ_{\mathbf{U}_{\mathcal{J}}} \in \{+_{\mathbf{U}_{\mathcal{J}}}, \times_{\mathbf{U}_{\mathcal{J}}}\}$ and $\diamond_{\mathbf{U}_{\mathcal{J}}} \in \{-_{\mathbf{U}_{\mathcal{J}}}, {}^{-1_{\mathbf{U}_{\mathcal{J}}}}\}$. Hereon, if the function type is apparent from the context, the subscripts $\mathcal{J}$ and $\mathbf{U}_{\mathcal{J}}$ will be omitted. For

instance, whenever unambiguous, we use the notations $\mathcal{Q}(u_1,\ldots,u_n)$ and $\mathcal{Q}(\mathsf{U}_1,\ldots,\mathsf{U}_n)$ for, respectively, a $\mathcal{J}$-function and its differentiation extension.

Here it will suffice to give an example. Let the $\mathcal{J}$-functions $u_1(S) = \cos S$ and $u_2(S) = S^3$ be both differentiable at some $S_0$, and let $\mathcal{Q}_{\mathcal{J}}(u_1, u_2)$ be differentiable at $S_0$ such that

$$\mathcal{Q}_{\mathcal{J}}(u_1, u_2) = u_1(S) + u_2(S) = \cos S + S^3.$$

The differentiation extension of $\mathcal{Q}_{\mathcal{J}}$ is then

$$\begin{aligned}
\mathcal{Q}_{\mathsf{U}_{\mathcal{J}}}(\mathsf{U}_1, \mathsf{U}_2) &= (u_1, \delta u_1)_{S_0} + (u_2, \delta u_2)_{S_0} \\
&= (\cos S, \delta \cos S)_{S_0} + (S^3, \delta S^3)_{S_0} \\
&= (\cos S + S^3, \delta(\cos S + S^3))_{S_0} \\
&= (\mathcal{Q}_{\mathcal{J}}(u_1, u_2), \delta \mathcal{Q}_{\mathcal{J}}(u_1, u_2))_{S_0}
\end{aligned}$$

By virtue of our definition of the *extension principle* for $\mathcal{J}$-functions (definition 6.1), we are able to define fundamental $\triangle_{\mathcal{J}}$-functions. For example, replacing $\mathcal{Q}$ by the "cos" function, one obtains the *trigonometric* $\triangle_{\mathcal{J}}$-function $\cos(u, \delta u)_{S_0} = (\cos(u), \delta \cos(u))_{S_0}$. In 'Machine implementation of interval auto-differentiation', we will give further discussion on differentiation extensions of $\mathcal{J}$-functions as well as more illustrative numerical examples.

Here, let us stress that restricting our discussion to single-variable $\mathcal{J}$-functions is not a loss of generality, since an $n$-variable $\mathcal{J}$-function can be viewed as a class of $n$ single-variable $\mathcal{J}$-functions. What is noteworthy in addition is that *higher-order interval auto-derivatives* can be computed in the framework of our system $\mathsf{T}_{\triangle_{\mathcal{J}}}$ of *dyadic* $\triangle_{\mathcal{J}}$-numbers ($\triangle_{\mathcal{J}}$-pairs). With the aid of definition 6.1, we next characterize the *n-differential operator* and the *n-differentiability predicate* for $\triangle_{\mathcal{J}}$-pairs.

**Definition 6.2** (*n*-**Differential Operator of a** $\triangle_{\mathcal{J}}$-**Number**): *For an integer* $n \geq 0$, *the n-differential operator of a* $\triangle_{\mathcal{J}}$-*pair* $\mathsf{U} = (u, \delta u) \in \mathsf{U}_{\mathcal{J}}$, *in symbols* $\delta^n \mathsf{U}$, *can be characterized recursively by*

(i) $\delta^0 \mathsf{U} = \mathsf{U}$,

(ii) $\delta^1 \mathsf{U} = (\delta u, \delta(\delta u)) = (\delta u, \delta^2 u) = \delta^1 \delta^0 \mathsf{U}$,

(iii) $n \geq 1 \Rightarrow \delta^n \mathsf{U} = (\delta^n u, \delta^{n+1} u) = \delta^1 \delta^{n-1} \mathsf{U}$.

**Definition 6.3** (*n*-**Differentiability Predicate for a** $\triangle_{\mathcal{J}}$-**Number**): *Let* $n \geq 0$. *The n-differentiability predicate for a* $\triangle_{\mathcal{J}}$-*pair* $\mathsf{U} = (u, \delta u) \in \mathsf{U}_{\mathcal{J}}$, *in symbols* $\mathrm{diff}^n(\mathsf{U})$, *is characterized by* $\mathrm{diff}^n(\mathsf{U}) \Leftrightarrow \delta^n \mathsf{U} \in \mathsf{U}_{\mathcal{J}}$.

Consequently, the next theorem, concerning *higher-order* auto-differentiability of $\mathcal{J}$-functions, is provable.

**Theorem 6.1** (*n*-**Differentiability Condition for a** $\triangle_{\mathcal{J}}$-**Number**): *Let* $n \geq 0$. *Then for a* $\triangle_{\mathcal{J}}$-*pair* $\mathsf{U} = (u, \delta u) \in \mathsf{U}_{\mathcal{J}}$, *we have* $\mathrm{diff}^n(\mathsf{U}) \Leftrightarrow \mathrm{diff}^{n+1}(u, S_0)$.

**Proof** It is clear that if the $\mathcal{J}$-function $u$ is $(n+1)$-differentiable at $S_0$, then, for $n \geq 0$, $(\delta^n u, \delta^{n+1} u)_{S_0} \in \mathsf{U}_{\mathcal{J}}$, and the proof follows by definition 6.3. $\square$

Our main objective in this section is to show that *higher-order* interval auto-derivatives are computable using only *dyadic* $\triangle_{\mathcal{J}}$-numbers ($\triangle_{\mathcal{J}}$-pairs). Towards this end, we need

readily available *Leibniz's rules* for $\Delta_{\mathcal{J}}$-numbers. By definitions 6.2 and 4.17, plus theorem 5.2, the following theorem is derivable.

**Theorem 23 (Leibniz's Rules for $\Delta_{\mathcal{J}}$-Numbers):** Let $\mathsf{Q}$ *and* $\mathsf{U}$ *be* $\Delta_{\mathcal{J}}$-*numbers. Then*

(i)  $\delta(\mathsf{Q}+\mathsf{U}) = \delta\mathsf{Q}+\delta\mathsf{U}$,

(ii)  $\delta(\mathsf{Q}\times\mathsf{U}) = \mathsf{Q}\times\delta\mathsf{U}+\mathsf{U}\times\delta\mathsf{Q}$.

By virtue of this theorem, and applying induction, the *general Leibniz rule* for $\Delta_{\mathcal{J}}$-numbers can be easily established. Let $\mathsf{Q}$ and $\mathsf{U}$ be $n$-times differentiable $\Delta_{\mathcal{J}}$-numbers. Then

$$\delta^n(\mathsf{QU}) = \sum_{k=0}^{n} \frac{n!}{k!(n-k)!} \left(\delta^{n-k}\mathsf{Q}\right)\left(\delta^{k}\mathsf{U}\right).$$

A nice consequence that we wish to point out is that with the *general Leibniz rule for $\Delta_{\mathcal{J}}$-numbers* at our disposal, and once we have in our machine implementation differentiation *kernels* (*seeds*) for the higher order dyads $(u, \delta u)$, $(\delta u, \delta^2 u)$, …, $(\delta^n u, \delta^{n+1} u)$, it is readily possible to compute *higher order* auto-derivatives by doing only *dyadic $\Delta_{\mathcal{J}}$-arithmetic*. In other words, within the framework of the theory $\mathsf{T}_{\Delta_{\mathcal{J}}}$ of *dyadic $\Delta_{\mathcal{J}}$-numbers*, higher order auto-differentiation is directly realizable without resorting to defining a *Grassmann algebra* for $n$-ary vectors of the form $(u, \delta u, \ldots, \delta^n u)$. This, along with some illustrative examples, will be discussed further in 'Machine implementation of interval auto-differentiation'.

Note also that considering only single-variable $\mathcal{J}$-functions is not a loss of generality, since an $n$-variable $\mathcal{J}$-function can be viewed as a class of $n$ single-variable $\mathcal{J}$-functions. Accordingly, *partial auto-derivatives*, *gradients* and *Hessians* are readily computable.

Finally, let us conclude this section with a few additional comments. In accord to definition 5.5, in the theory $\mathsf{T}_{\Delta_{\mathcal{J}}}$ of $\Delta_{\mathcal{J}}$-numbers, a *singleton* $\Delta_{\mathcal{J}}$-number defines a $\Delta_{\mathbb{R}}$-number. That is, all the results of this section apply to $\Delta_{\mathbb{R}}$-arithmetic as well. Moreover, computing the $\Delta_{\mathcal{J}}$-number $(q, \delta q)_{S_0}$ is very useful in practice. In engineering and physical sciences, a recurring problem is to compute the derivatives under parametric uncertainty (For further details, the reader may consult, *e.g.*, *Dawood, 2014*, *Dawood & Dawood, 2022*; *Kulisch, 2013*; *Moore, 1966*, *Neidinger, 2010*; *Sommer, Pradalier & Furgale, 2016*, and *Tingelstad & Egeland, 2017*). Also noteworthy here is that with a few basic modifications, the categorical system $\mathsf{T}_{\Delta_{\mathcal{J}}}$ axiomatized in this text can be extended analogously to compute *fuzzy auto-derivatives* (For further details on fuzzy analysis, see, *e.g.*, *Goetschel & Voxman, 1986* and *Puri & Ralescu, 1983*).

## THE ALGEBRAIC STRUCTURE OF INTERVAL DIFFERENTIATION ARITHMETIC

Building on the parts of the theory established in 'A differential interval algebra' and 'A categorical axiomatization of interval differentiation arithmetic', this section provides a detailed investigation of the algebraic structure of $\Delta_{\mathcal{J}}$-numbers. By virtue of the categoricity of the theory $\mathsf{T}_{\Delta_{\mathcal{J}}}$ (theorem 5.4), the properties of $\mathcal{J}$-numbers are assumed priori.

We commence this section by establishing the algebraic properties of $\triangle_{\mathcal{J}}$-addition and $\triangle_{\mathcal{J}}$-multiplication.

**Theorem 7.1 (Algebraic Properties of $\triangle_{\mathcal{J}}$-Addition):** *The following algebraic properties hold for $\triangle_{\mathcal{J}}$-addition.*

(i) *Identity element for* $+.$ $\left(\forall Q \in \mathbf{U}_{\mathcal{J}}\right)\left(0_{\mathbf{U}_{\mathcal{J}}} + Q = Q + 0_{\mathbf{U}_{\mathcal{J}}} = Q\right),$

(ii) *Inverses for* $+.$ $\left(\forall Q, U \in \mathbf{U}_{\mathcal{J}}\right)\left(Q + U = 0_{\mathbf{U}_{\mathcal{J}}} \Leftrightarrow Q \in \mathbf{U}_{[q]} \wedge U = -Q\right),$

(iii) *Cancellativity for* $+.$ $\left(\forall Q, U, V \in \mathbf{U}_{\mathcal{J}}\right)\left(Q + V = U + V \Rightarrow Q = U\right),$

(iv) *Commutativity for* $+.$ $\left(\forall Q, U \in \mathbf{U}_{\mathcal{J}}\right)\left(Q + U = U + Q\right),$

(v) *Associativity for* $+.$ $\left(\forall Q, U, V \in \mathbf{U}_{\mathcal{J}}\right)\left(Q + (U + V) = (Q + U) + V\right).$

**Proof** The proof for (i) follows from theorem 5.2. (ii) follows from theorems 4.6 and 5.2. By cancellativity, commutativity and associativity of $\mathcal{J}$-addition, (iii), (iv) and (v) are easily provable by theorem 5.2. $\square$

**Theorem 7.2 (Algebraic Properties of $\triangle_{\mathcal{J}}$-Multiplication):** *The following algebraic properties hold for $\triangle_{\mathcal{J}}$-multiplication.*

(i) *Annihilating element for* $\times.$ $\left(\forall Q \in \mathbf{U}_{\mathcal{J}}\right)\left(0_{\mathbf{U}_{\mathcal{J}}} \times Q = Q \times 0_{\mathbf{U}_{\mathcal{J}}} = 0_{\mathbf{U}_{\mathcal{J}}}\right),$

(ii) *Identity element for* $\times.$ $\left(\forall Q \in \mathbf{U}_{\mathcal{J}}\right)\left(1_{\mathbf{U}_{\mathcal{J}}} \times Q = Q \times 1_{\mathbf{U}_{\mathcal{J}}} = Q\right),$

(iii) *Inverses for* $\times.$ $\left(\forall Q, U \in \mathbf{U}_{\mathcal{J}}\right)\left(Q \times U = 1_{\mathbf{U}_{\mathcal{J}}} \Leftrightarrow Q \in \mathbf{U}_{[q]} \wedge U = Q^{-1} \wedge 0 \notin q(S_0)\right),$

(iv) *Cancellativity for* $\times.$ $\left(\forall Q, U, V \in \mathbf{U}_{\mathcal{J}}\right)\left((Q \times V = U \times V \Rightarrow Q = U) \Leftrightarrow 0 \notin v(S_0)\right),$

(v) *Commutativity for* $\times.$ $\left(\forall Q, U \in \mathbf{U}_{\mathcal{J}}\right)\left(Q \times U = U \times Q\right).$

**Proof** The proof for (i) and (ii) follows immediately from theorem 5.2. For (iii), assume that $Q \times U = 1_{\mathbf{U}_{\mathcal{J}}} = ([1], [0])$, which yields, by theorems 5.2, 4.6, and the invertibility properties of $\mathcal{J}$-arithmetic, that $Q \in \mathbf{U}_{[q]} \wedge U = Q^{-1} \wedge 0 \notin q(S_0)$. The converse direction is easily derivable by assuming the right hand side. By the cancellative properties of $\mathcal{J}$-arithmetic, (iv) follows from theorems 5.2 and 4.2. By commutativity of $\mathcal{J}$-multiplication, (v) is easily derivable from theorem 5.2. $\square$

Thus, not all elements of $\mathbf{U}_{\mathcal{J}}$ are invertible for addition or multiplication. A $\triangle_{\mathcal{J}}$-number $Q$ is invertible for addition if, and only if, it is a point $\triangle_{\mathcal{J}}$-number and is invertible for multiplication if, and only if, it is a point $\triangle_{\mathcal{J}}$-number with $0 \notin q(S_0)$. Also, unlike interval arithmetic, $\triangle_{\mathcal{J}}$-arithmetic has nonzero *zero divisors*, since $\left([0], [\underline{\alpha}, \overline{\alpha}]\right) \times \left([0], [\underline{\beta}, \overline{\beta}]\right) = 0_{\mathbf{U}_{\mathcal{J}}}$. Moreover, $\triangle_{\mathcal{J}}$-multiplication is not associative, which is figured in the following theorem.

**Theorem 7.3 (Associativity of $\triangle_{\mathcal{J}}$-Multiplication):** *In general, $\triangle_{\mathcal{J}}$-multiplication is not associative. That is*

$$\left(\exists Q, U, V \in \mathbf{U}_{\mathcal{J}}\right)\left(Q \times (U \times V) \neq (Q \times U) \times V\right).$$

**Proof** We prove the statement by a counter example. Let $q$, $u$, and $v$ be $\mathcal{J}$-functions defined respectively as $q(S) = S^2 + S$, $u(S) = S$, and $v(S) = [3] \times S^2$. Then, for $S_0 = [-1, 1]$, we have the corresponding $\triangle_{\mathcal{J}}$-numbers $Q = (q, \delta q) = ([-1, 2], [-1, 3])$, $U = (u, \delta u) = ([-1, 1], [1])$, and $V = (v, \delta v) = ([0, 3], [-6, 6])$. Now, by theorem 5.2, we have

$Q \times (U \times V) = ([-6,6],[-21,27])$ and $(Q \times U) \times V = ([-6,6],[-24,27])$. Hence $Q \times (U \times V) \neq (Q \times U) \times V$. □

However not associative, $\triangle_{\mathcal{J}}$-multiplication satisfies two weak associativity properties namely *subalternativity* and *flexibility*. These are established in the next two theorems.

**Theorem 7.4 (Subalternativity of $\triangle_{\mathcal{J}}$-Multiplication):** $\triangle_{\mathcal{J}}$-*multiplication is subalternative, that is*

$$\left(\forall Q, U \in \mathbf{U}_{\mathcal{J}}\right)\left(Q \times (Q \times U) \subseteq (Q \times Q) \times U\right).$$

**Proof** Let $Q$ and $U$ be in $\mathbf{U}_{\mathcal{J}}$. Then, from the associative and subdistributive properties of $\mathcal{J}$-arithmetic, we have

$$
\begin{aligned}
Q \times (Q \times U) &= \left(q \times (q \times u), \delta q \times (q \times u) + q \times (\delta q \times u + q \times \delta u)\right) \\
&= \left((q \times q) \times u, (\delta q \times q) \times u + q \times (\delta q \times u + q \times \delta u)\right) \\
&\subseteq \left((q \times q) \times u, (\delta q \times q) \times u + (q \times \delta q) \times u + (q \times q) \times \delta u\right) \\
&= \left((q \times q) \times u, [2] \times (q \times \delta q) \times u + (q \times q) \times \delta u\right) \\
&= \left(q \times q, [2] \times (q \times \delta q)\right) \times (u, \delta u) \\
&= \left((q, \delta q) \times (q, \delta q)\right) \times (u, \delta u) \\
&= (Q \times Q) \times U.
\end{aligned}
$$

Therefore, multiplication is subalternative in $\mathbf{U}_{\mathcal{J}}$. □

**Theorem 7.5 (Flexibility of $\triangle_{\mathcal{J}}$-Multiplication):** $\triangle_{\mathcal{J}}$-*multiplication is flexible, that is*

$$\left(\forall Q, U \in \mathbf{U}_{\mathcal{J}}\right)\left((Q \times U) \times Q = Q \times (U \times Q)\right).$$

**Proof** The theorem follows by the fact that $\triangle_{\mathcal{J}}$-multiplication is commutative (theorem 7.2). □

Now, we turn to the algebraic property of distributivity. Like the case with $\mathcal{J}$-arithmetic, $\triangle_{\mathcal{J}}$-multiplication is not distributive over $\triangle_{\mathcal{J}}$-addition. For example, consider the three $\triangle_{\mathcal{J}}$-numbers $Q$, $U$, and $V$ given in the proof of theorem 7.3. Then we have $Q \times (U + V) = ([-4,8],[-14,26])$, while $Q \times U + Q \times V = ([-5,8],[-19,26])$, and hence $Q \times (U + V) \neq Q \times U + Q \times V$. In contrast, $\triangle_{\mathcal{J}}$-arithmetic is *subdistributive* ($\mathsf{S}$-*distributive*). This is established in the next theorem.

**Theorem 7.6 (Subdistributivity in $\triangle_{\mathcal{J}}$-Numbers):** $\triangle_{\mathcal{J}}$-*arithmetic is subdistributive, that is*

$$\left(\forall Q, U, V \in \mathbf{U}_{\mathcal{J}}\right)\left(V \times (Q + U) \subseteq V \times Q + V \times U\right).$$

**Proof** Let $Q$, $U$, and $V$ be any three $\Delta_{\mathcal{J}}$-numbers. According to theorem 5.2, and assuming the properties of interval operations, we have

$$
\begin{aligned}
V \times (Q+U) &= \left(v \times (q+u), \delta v \times (q+u) + v \times (\delta q + \delta u)\right) \\
&\subseteq \left(v \times q + v \times u, \delta v \times q + \delta v \times u + v \times \delta q + v \times \delta u\right) \\
&= \left(v \times q + v \times u, (\delta v \times q + v \times \delta q) + (\delta v \times u + v \times \delta u)\right) \\
&= \left(v \times q, (\delta v \times q + v \times \delta q)\right) + \left(v \times u, (\delta v \times u + v \times \delta u)\right) \\
&= V \times Q + V \times U,
\end{aligned}
$$

and therefore multiplication *subdistributes* over addition in $\mathbf{U}_{\mathcal{J}}$. $\qquad\square$

The preceding theorem establishes left $S$-distributivity. Right $S$-distributivity follows from commutativity of $\Delta_{\mathcal{J}}$-multiplication.

We will now make use of the previous results to prove the next theorem about the algebraic structure of $\Delta_{\mathcal{J}}$-numbers.

**Theorem 7.7 (Commutative NA $S$-Semiring of $\Delta_{\mathcal{J}}$-Numbers):** *The structure* $\mathfrak{U}_{\mathcal{J}} = \langle \mathbf{U}_{\mathcal{J}}; +_{\mathbf{U}_{\mathcal{J}}}, \times_{\mathbf{U}_{\mathcal{J}}}; 0_{\mathbf{U}_{\mathcal{J}}}, 1_{\mathbf{U}_{\mathcal{J}}} \rangle$ *is a commutative* $\times$-NA $S$-semiring in which $\times_{\mathbf{U}_{\mathcal{J}}}$ *is subalternative and flexible.*

**Proof** By theorem 7.1, the additive structure $\langle \mathbf{U}_{\mathcal{J}}; +_{\mathbf{U}_{\mathcal{J}}}; 0_{\mathbf{U}_{\mathcal{J}}} \rangle$ is a cancellative commutative monoid. By theorems 7.2 and 7.3, the multiplicative structure $\langle \mathbf{U}_{\mathcal{J}}; \times_{\mathbf{U}_{\mathcal{J}}}; 1_{\mathbf{U}_{\mathcal{J}}} \rangle$ is a noncancellative commutative NA-monoid. In consequence of theorem 7.6, $\times_{\mathbf{U}_{\mathcal{J}}}$ subdistributes over $+_{\mathbf{U}_{\mathcal{J}}}$. By theorem 7.2, $0_{\mathbf{U}_{\mathcal{J}}}$ is an absorbing element for $\times_{\mathbf{U}_{\mathcal{J}}}$. According to definition 2.10, the structure $\mathfrak{U}_{\mathcal{J}}$, of $\Delta_{\mathcal{J}}$-arithmetic, is therefore a commutative $\times$-NA $S$-semiring. Lastly, by theorems 7.4 and 7.5, $\times_{\mathbf{U}_{\mathcal{J}}}$ is subalternative and flexible. $\qquad\square$

Lastly, we prove two special results on the structure $\mathfrak{U}_{[q]}$ of point $\Delta_{\mathcal{J}}$-numbers.

**Theorem 7.8 (Sub-Algebraicity of Point $\Delta_{\mathcal{J}}$-numbers):** *The structure* $\mathfrak{U}_{[q]}$ *of point* $\Delta_{\mathcal{J}}$-numbers is a subalgebra of the structure $\mathfrak{U}_{\mathcal{J}}$ of $\Delta_{\mathcal{J}}$-numbers. In symbols $\mathfrak{U}_{[q]} \sqsubseteq \mathfrak{U}_{\mathcal{J}}$.

**Proof** By definition of $\Delta_{\mathcal{J}}$-numbers, $0_{\mathbf{U}_{\mathcal{J}}}$ and $1_{\mathbf{U}_{\mathcal{J}}}$ are both elements of the set $\mathbf{U}_{[q]} \subset \mathbf{U}_{\mathcal{J}}$. By theorems 4.6 and 5.1, for $\circ \in \{+_{\mathbf{U}_{\mathcal{J}}}, \times_{\mathbf{U}_{\mathcal{J}}}\}$ and for any $[q]$ and $[u]$ in $\mathbf{U}_{[q]}$, $[q] \circ [u]$ is in turn in $\mathbf{U}_{[q]}$. Then, the criteria for the subalgebraicity of $\mathfrak{U}_{[q]}$ is established and therefore $\mathfrak{U}_{[q]} \sqsubseteq \mathfrak{U}_{\mathcal{J}}$. $\qquad\square$

**Theorem 7.9 (Commutative Ring of Point $\Delta_{\mathcal{J}}$-Numbers):** *The structure* $\mathfrak{U}_{[q]} = \langle \mathbf{U}_{[q]}; +_{\mathbf{U}_{\mathcal{J}}}, \times_{\mathbf{U}_{\mathcal{J}}}; 0_{\mathbf{U}_{\mathcal{J}}}, 1_{\mathbf{U}_{\mathcal{J}}} \rangle$ *is a commutative unital ring in which every element whose first component is nonzero has an inverse for* $\times_{\mathbf{U}_{\mathcal{J}}}$.

**Proof** Restricting the operations $+_{\mathbf{U}_{\mathcal{J}}}$ and $\times_{\mathbf{U}_{\mathcal{J}}}$ to the set $\mathbf{U}_{[q]}$ in theorems 7.3, 7.6, and 7.2, it follows respectively that $\times_{\mathbf{U}_{\mathcal{J}}}$ is associative, $\times_{\mathbf{U}_{\mathcal{J}}}$ distributes over $+_{\mathbf{U}_{\mathcal{J}}}$, and every element whose first component is nonzero has a $\times_{\mathbf{U}_{\mathcal{J}}}$-inverse in $\mathbf{U}_{[q]}$. The proof is thus established in consequence of theorem 7.7. $\qquad\square$

That is, the multiplicatively non-associative $S$-semiring $\mathfrak{U}_{\mathcal{J}}$ of $\Delta_{\mathcal{J}}$-numbers has as a subalgebra a commutative unital ring $\mathfrak{U}_{[q]}$.

# MONOTONICITY AND ISOMORPHISM THEOREMS FOR INTERVAL DIFFERENTIATION NUMBERS

In this section, some monotonicity and isomorphism theorems for $\Delta_{\mathcal{J}}$-numbers are established, and finally a corollary concerning the structure of $\Delta_{\mathbb{R}}$-numbers is entailed. A first key result we will next prove is the inclusion monotonicity theorem for $\Delta_{\mathcal{J}}$-arithmetic, which establishes that the inclusion relation is compatible with the algebraic $\Delta_{\mathcal{J}}$-operations.

**Theorem 8.1 (Inclusion Monotonicity for $\Delta_{\mathcal{J}}$-Numbers):** *Let $\mathsf{Q}_1$, $\mathsf{Q}_2$, $\mathsf{U}_1$, and $\mathsf{U}_2$ be $\Delta_{\mathcal{J}}$-numbers such that $\mathsf{Q}_1 \subseteq \mathsf{U}_1$ and $\mathsf{Q}_2 \subseteq \mathsf{U}_2$. Let $\circ \in \{+, \times\}$ be a binary $\Delta_{\mathcal{J}}$-operation and $\diamond \in \{-, ^{-1}\}$ be a definable unary $\Delta_{\mathcal{J}}$-operation. Then*
(i) $\mathsf{Q}_1 \circ \mathsf{Q}_2 \subseteq \mathsf{U}_1 \circ \mathsf{U}_2$,
(ii) $\diamond \mathsf{Q}_1 \subseteq \diamond \mathsf{U}_1$.

**Proof** By hypothesis, we have $\mathsf{Q}_1 \subseteq \mathsf{U}_1$ and $\mathsf{Q}_2 \subseteq \mathsf{U}_2$. Then, according to definition 5.3 and theorem 4.4, we have

$$
\begin{aligned}
\mathsf{Q}_1 + \mathsf{Q}_2 &= (q_1, \delta q_1) + (q_2, \delta q_2) \\
&= (q_1 + q_2, \delta q_1 + \delta q_2) \\
&\subseteq (u_1 + u_2, \delta u_1 + \delta u_2) \\
&= \mathsf{U}_1 + \mathsf{U}_2.
\end{aligned}
$$

Analogously, $\mathsf{Q}_1 \times \mathsf{Q}_2 \subseteq \mathsf{U}_1 \times \mathsf{U}_2$ and $\diamond \mathsf{Q}_1 \subseteq \diamond \mathsf{U}_1$. This completes the proof. $\square$

In consequence of this theorem, from the fact that $([q(s_0)], [\delta q(s_0)]) \subseteq \mathsf{Q} \Leftrightarrow \mathsf{q} \in Q$, we have the following important special case.

**Corollary 8.1 (Membership Monotonicity for $\Delta_{\mathcal{J}}$-Numbers):** *Let $\mathsf{Q}$ and $\mathsf{U}$ be $\Delta_{\mathcal{J}}$-numbers with $\mathsf{q} \in Q$ and $\mathsf{u} \in U$. Let $\circ \in \{+, \times\}$ be a binary $\Delta_{\mathcal{J}}$-operation and $\diamond \in \{-, ^{-1}\}$ be a definable unary $\Delta_{\mathcal{J}}$-operation. Then*
(i) $\mathsf{q} \circ \mathsf{u} \in Q \circ U$,
(ii) $\diamond \mathsf{q} \in \diamond \mathsf{Q}$.

Two important results, concerning *isomorphism theorems* for $\Delta_{\mathcal{J}}$-arithmetic, are figured in the following theorems.

**Theorem 8.2 (Isomorphicity to $\Delta_{\mathbb{R}}$-Numbers):** *The structure $\mathfrak{U}_{[\mathsf{q}]} = \langle \mathbf{U}_{[\mathsf{q}]}; +_{\mathbf{U}_{\mathcal{J}}}, \times_{\mathbf{U}_{\mathcal{J}}} \rangle$ is isomorphic to the structure $\mathfrak{U}_{\mathbb{R}} = \langle \mathbf{U}_{\mathbb{R}}; +_{\mathbf{U}_{\mathbb{R}}}, \times_{\mathbf{U}_{\mathbb{R}}} \rangle$ of $\Delta_{\mathbb{R}}$-numbers. In symbols $\mathfrak{U}_{[\mathsf{q}]} \simeq \mathfrak{U}_{\mathbb{R}}$.*

**Proof** Let $\iota : \mathbf{U}_{\mathbb{R}} \hookrightarrow \mathbf{U}_{[\mathsf{q}]}$ be the mapping from $\mathbf{U}_{\mathbb{R}}$ to $\mathbf{U}_{[\mathsf{q}]}$ given by

$$
\iota(\mathsf{q}) = [\mathsf{q}] = ([q], [\delta q]).
$$

By means of the fact that point intervals are isomorphic to real numbers (theorem 4.5), it is straightforward to show that $\iota$ is an isomorphism from $\mathbf{U}_{\mathbb{R}}$ onto $\mathbf{U}_{[\mathsf{q}]}$. $\square$

**Theorem 8.3 (Isomorphicity to $\mathcal{J}$-Numbers):** *The structure $\mathfrak{U}_{(\mathcal{J}, 0)} = \langle \mathbf{U}_{(\mathcal{J}, 0)}; +_{\mathbf{U}_{\mathcal{J}}}, \times_{\mathbf{U}_{\mathcal{J}}} \rangle$ is isomorphic to the commutative $\mathsf{S}$-semiring $\mathfrak{J}_{\mathbb{R}} = \langle \mathcal{J}_{\mathbb{R}}; +_{\mathcal{J}}, \times_{\mathcal{J}} \rangle$ of interval numbers.*

**Proof** Let $\iota : \mathcal{J}_{\mathbb{R}} \hookrightarrow \mathbf{U}_{(\mathcal{J},0)}$ be the mapping from $\mathcal{J}_{\mathbb{R}}$ to $\mathbf{U}_{(\mathcal{J},0)}$ defined by $\iota\left(\left[\underline{\alpha}, \overline{\alpha}\right]\right) = \left(\left[\underline{\alpha}, \overline{\alpha}\right], [0]\right)$. Obviously, $\iota$ is an isomorphism from $\mathcal{J}_{\mathbb{R}}$ onto $\mathbf{U}_{(\mathcal{J},0)}$ and therefore $\mathfrak{U}_{(\mathcal{J},0)} \simeq \mathfrak{J}_{\mathbb{R}}$. □

Accordingly, up to isomorphism, the sets $\mathcal{J}_{\mathbb{R}}$ and $\mathbf{U}_{(\mathcal{J},0)}$ are equivalent, and therefore the structure $\mathfrak{U}_{(\mathcal{J},0)}$ is a commutative S-semiring.

In consequence of theorems 7.9 and 8.2, we have the following corollary concerning the structure of $\Delta_{\mathbb{R}}$-numbers.

**Corollary 8.2 (Commutative Ring of $\Delta_{\mathbb{R}}$-Numbers):** *The algebra* $\mathfrak{U}_{\mathbb{R}} = \langle \mathbf{U}_{\mathbb{R}}; +_{\mathbf{U}_{\mathbb{R}}}, \times_{\mathbf{U}_{\mathbb{R}}}; 0_{\mathbf{U}_{\mathbb{R}}}, 1_{\mathbf{U}_{\mathbb{R}}} \rangle$ *of* $\Delta_{\mathbb{R}}$*-numbers is a commutative unital ring in which every element whose first component is nonzero has an inverse for* $\times_{\mathbf{U}_{\mathbb{R}}}$.

## MACHINE IMPLEMENTATION OF INTERVAL AUTO-DIFFERENTIATION

In this last section, we consider some aspects of the computational implementation of interval auto-differentiation in the framework of our theory $\mathsf{T}_{\Delta_{\mathcal{J}}}$ of $\Delta_{\mathcal{J}}$-numbers ($\Delta_{\mathcal{J}}$-pairs). The algorithm of the theory is coded in Common Lisp as a part of the software package InCLosure (InCL) (*Dawood, 2020* and *Dawood, 2023*). After providing a *mathematical* flavor of the algorithm,[9] we offer insights of the theory by giving some simple examples that illustrate how to concurrently compute guaranteed enclosures of images of families of real functions and their derivatives, then we deal with a more sophisticated problem whose result values will be calculated to an arbitrary precision using InCL commands, and finally, we give a brief account of how to calculate higher order interval auto-derivatives using the theory $\mathsf{T}_{\Delta_{\mathcal{J}}}$ of *dyadic* $\Delta_{\mathcal{J}}$-arithmetic.

In a way analogous to that of $\Delta_{\mathbb{R}}$-arithmetic (see, *e.g.*, *Dawood, 2014* and *Dawood & Megahed, 2019*), $\Delta_{\mathcal{J}}$-arithmetic can be machine realized. Toward calculating the $\Delta_{\mathcal{J}}$-pair of a differentiable $\mathcal{J}$-function at $S_0 \in \mathcal{J}$, we begin with a *minimal class* of symbolic rules of differentiable $\mathcal{J}$-functions and their derivatives which acts as $\Delta_{\mathcal{J}}$*-seeds* ($\Delta_{\mathcal{J}}$*-kernels*) for carrying out the computation. As examples of $\Delta_{\mathcal{J}}$-kernels, one can start with the following *elementary* $\Delta_{\mathcal{J}}$-pairs.

$\left(AS^b, AbS^{b-1}\right)$, $(\ln(S), 1/S)$, $\left(e^S, e^S\right)$, $(\sin(S), \cos(S))$, $(\cos(S), -\sin(S))$, and so forth.

The class of *unary* $\Delta_{\mathcal{J}}$*-kernels* will be denoted by $\mathbf{P}_{\langle 1 \rangle}$, and we will understand by $\mathcal{J}$*-kernels* a class $\mathbf{K}_{\langle 1 \rangle} = \left\{q \in \mathcal{J}_{\langle 1 \rangle} \mid (q, \delta q) \in \mathbf{P}_{\langle 1 \rangle}\right\}$. Accordingly, the *first-order* interval auto-derivative of a $\mathcal{J}$-function $q(S)$, at $S_0 \in \mathcal{J}$, can be viewed as

$$q(S), S_0 \overset{\text{Input}}{\Longrightarrow} \begin{bmatrix} \Delta_{\mathcal{J}}\text{-Kernels} \\ \text{Chain Rule} \\ \Delta_{\mathcal{J}}\text{-Algebra} \end{bmatrix} \overset{\text{Output}}{\Longrightarrow} (q, \delta q)_{S_0}.$$

To further illustrate, we next give some examples that can be worked by hand.

**Example 9.1 ($\Delta_{\mathcal{J}}$-Number for the Cosine Function):** *Consider the $\mathcal{J}$-function*

$$q(S) = \cos\left(\sqrt{S}\right) \text{ with } \underline{s} \geq 0.$$

[9]The two *types* (*modes*) of algorithmic differentiation are both realizable in the theory $\mathsf{T}_{\Delta_{\mathcal{J}}}$ of $\Delta_{\mathcal{J}}$-algebra. Here, we consider only the *forward-type*. With a few basic modifications, the *reversed-type* is implementable as well.

*Computing the* $\Delta_{\mathcal{J}}$-*pair* $(q, \delta q)_{[1,4]}$ *yields*

$$(q, \delta q)_{[1,4]} = \left(\cos\left(\sqrt{[1,4]}\right), -\left(\sin\left(\sqrt{[1,4]}\right)\right)/2\sqrt{[1,4]}\right)_{[1,4]}$$

$$= \left(\cos\left(\left[\sqrt{1}, \sqrt{4}\right]\right), -\left(\sin\left(\left[\sqrt{1}, \sqrt{4}\right]\right)\right)/2\left[\sqrt{1}, \sqrt{4}\right]\right)_{[1,4]}$$

$$= (\cos([1,2]), -(\sin([1,2]))/2[1,2])_{[1,4]}$$

$$= ([\cos(2), \cos(1)], -[\sin(1), 1]/[2,4])_{[1,4]}$$

$$= ([\cos(2), \cos(1)], [-1/2, -1/4][\sin(1), 1])_{[1,4]}$$

$$= ([\cos(2), \cos(1)], [-1/2, (-1/4)\sin(1)])_{[1,4]}.$$

The first component of the resulting $\Delta_{\mathcal{J}}$-pair, $[\cos(2), \cos(1)]$, is a *guaranteed enclosure* of the image of the real function $q(s) = \cos\left(\sqrt{s}\right)$ over the interval $[1,4]$, while the second component, $[-1/2, (-1/4)\sin(1)]$, is a *guaranteed enclosure* of the image of the real function $\delta q(s)$ over the same interval. For example,

$$q(4) = \cos(2) \in [\cos(2), \cos(1)] = q([1,4]),$$

$$\delta q(4) = -\sin(2)/4 \in [-1/2, (-1/4)\sin(1)] = \delta q([1,4]).$$

**Example 9.2** ($\Delta_{\mathcal{J}}$-**Number for a Family of Real Functions**): *Let* $\mathbf{Q}_{\mathbb{R}}$ *be the family of real functions*

$$q_{\mathbb{R}}(s, a, b, c) = as^3 + bs^2 + c,$$

*where the variable* $s \in S = [-1, 2]$ *and the constants* $a$, $b$, *and* $c$ *are respectively in* $[-1, 1]$, $[0, 1]$, *and* $[0, 2]$. *We want to compute enclosures of the images of the family* $\mathbf{Q}_{\mathbb{R}}$ *and its derivative* $\delta\mathbf{Q}_{\mathbb{R}}$. *The interval extension of* $\mathbf{Q}_{\mathbb{R}}$ *is the interval function*

$$q(S, [-1,1], [0,1], [0,2]) = [-1,1]S^3 + [0,1]S^2 + [0,2].$$

*The* $\Delta_{\mathcal{J}}$-*pair* $(q, \delta q)_{[-1,2]}$ *at the interval* $S_0 = [-1, 2]$ *is computed as follows.*

$$(q, \delta q)_{[-1,2]}$$

$$= \left([-1,1]([-1,2])^3 + [0,1]([-1,2])^2 + [0,2], 3[-1,1]([-1,2])^2 + 2[0,1][-1,2]\right)_{[-1,2]}$$

$$= ([-1,1][-1,8] + [0,1][0,4] + [0,2], [-3,3][0,4] + [0,2][-1,2])_{[-1,2]}$$

$$= ([-8,8] + [0,4] + [0,2], [-12,12] + [-2,4])_{[-1,2]}$$

$$= ([-8,14], [-14,16])_{[-1,2]}.$$

*The intervals* $[-8,14]$ *and* $[-14,16]$ *are guaranteed enclosures of the images of the family* $\mathbf{Q}_{\mathbb{R}}$ *and its derivative* $\delta\mathbf{Q}_{\mathbb{R}}$ *over the interval* $[-1,2]$ *respectively. In fact, the interval* $[-8,14]$ *is the exact image,* $I_{\mathbf{Q}}$, *of* $\mathbf{Q}_{\mathbb{R}}$ *over the interval* $[-1,2]$.

The previous example clearly embodies that the theory $\mathsf{T}_{\Delta_{\mathcal{J}}}$ presented in this text is powerful and reliable for simultaneously providing guaranteed enclosures of families of real functions and their derivatives. For instance, the following real functions and their derivatives are members of the families $\mathbf{Q}_{\mathbb{R}}$ and $\delta\mathbf{Q}_{\mathbb{R}}$ of example 9.2 respectively.

$$q_1(s) = s^3 + s^2 + 2, q_2(s) = -s^3$$

$\delta q_1(s) = 3s^2 + 2s, \ \delta q_2(s) = -3s^2.$

The exact images of these functions are included in the result of example 9.2 as follows.

$I_{q_1}([-1,2]) = [2,14] \subseteq [-8,14] = q([-1,2]),$

$I_{q_2}([-1,2]) = [-8,1] \subseteq [-8,14] = q([-1,2]),$

$I_{\delta q_1}([-1,2]) = \left[-\dfrac{1}{3}, 16\right] \subseteq [-14,16] = \delta q([-1,2]),$

$I_{\delta q_2}([-1,2]) = [-12,0] \subseteq [-14,16] = \delta q([-1,2]).$

The overestimation of the exact image of $\delta \mathbf{Q}_{\mathbb{R}}$ in example 9.2 naturally arises from the *interval dependency problem*. One noteworthy virtue of the subdivision theorem for $\Delta_{\mathcal{J}}$-numbers (theorem 5.7) is that one can deploy the *subdivision method* to decrease the overestimation and hence obtain *arbitrarily* sharper intervals that get closer to the exact image. The next example clarifies the matters.

**Example 9.3 ($\Delta_{\mathcal{J}}$-Number with Subdivision):** *Consider again the $\mathcal{J}$-function of example 9.2 given by*

$q(S, [-1,1], [0,1], [0,2]) = [-1,1]S^3 + [0,1]S^2 + [0,2].$

*We desire to compute the $\Delta_{\mathcal{J}}$-pair $(q, \delta q)_{[-1,2]}$ at the interval $S_0 = [-1,2]$ by subdividing the interval $[-1,2]$ into the three subintervals $[-1,0]$, $[0,1]$, and $[1,2]$ of width 1. Then, we compute the $\Delta_{\mathcal{J}}$-numbers $(q, \delta q)_{[-1,0]}$, $(q, \delta q)_{[0,1]}$, and $(q, \delta q)_{[1,2]}$ as follows.*

$(q, \delta q)_{[-1,0]}$

$= \left([-1,1]([-1,0])^3 + [0,1]([-1,0])^2 + [0,2], 3[-1,1]([-1,0])^2 + 2[0,1]([-1,0])\right)_{[-1,0]}$

$= ([-1,1][-1,0] + [0,1][0,1] + [0,2], [-3,3][0,1] + [0,2][-1,0])_{[-1,0]}$

$= ([-1,1] + [0,1] + [0,2], [-3,3] + [-2,0])_{[-1,0]}$

$= ([-1,4], [-5,3])_{[-1,0]},$

*Similarly, $(q, \delta q)_{[0,1]} = ([-1,4], [-3,5])_{[0,1]}$ and $(q, \delta q)_{[1,2]} = ([-8,14], [-12,16])_{[1,2]}$. Hence the resulting $\Delta_{\mathcal{J}}$-number by the subdivision technique is given by*

$([-1,4] \cup [-1,4] \cup [-8,14], [-5,3] \cup [-3,5] \cup [-12,16]) = ([-8,14], [-12,16]).$

*Compared to the naive result in example 9.2, the resulting $\Delta_{\mathcal{J}}$-pair with the subdivision method gives a better enclosure, fortunately it is the exact result, for the image of the first derivative. That is*

$([-8,14], [-12,16]) \subseteq ([-8,14], [-14,16]).$

In the previous example, the subdivision technique with only three subintervals yields the exact images. In some problems, when the result is far away from the exact images, increasing the number of subintervals could give arbitrarily better enclosures of the images but with the disadvantage of increased computational time. Now, we move on to compute,

**Table 1  InCLosure values for interval automatic differentiation.**

| Subdivisions | InCLosure values for the pair $(q, \delta q)_{[-1,2]}$ |
|---|---|
| 1 (no subdivision) | $\left( \begin{array}{c} [-515.0, 191.4427985505345723919], \\ [-147294.44809204868892190371, 145297.44809204868892190371] \end{array} \right)$ |
| 5 | $\left( \begin{array}{c} [-499.940928, 179.8726123514945723919], \\ [-77358.50282798795561838428, 75356.39752085195561838428] \end{array} \right)$ |
| 10 | $\left( \begin{array}{c} [-466.724862, 110.8030425325745723919], \\ [-75987.53606834857190112981, 73699.56740236057190112981] \end{array} \right)$ |
| 20 | $\left( \begin{array}{c} [-434.82104971875, 39.8387755903470333294], \\ [-75400.15661633499832211557, 72660.87036293595925961557] \end{array} \right)$ |
| 50 | $\left( \begin{array}{c} [-408.379583369088, 19.78286094370217251581], \\ [-75067.10983026439761984353, 71855.91048800864805542753] \end{array} \right)$ |
| 100 | $\left( \begin{array}{c} [-398.096543381742, 14.8675821888729383479], \\ [-74958.36726177567900543373, 71542.87733701227696496573] \end{array} \right)$ |

to an *arbitrarily* sharper intervals, the result of a more sophisticated example using the software package InCLosure.

**Example 9.4 (Interval Auto-Differentiation using InCLosure):**  *Let $q(S)$ be a $\mathcal{J}$-function defined by*

$$q(S) = [-0.5, -0.4] S^{10} - [-3, -2] S^6 - \sin\left( e^{\left( \frac{5}{8} \left( \sin\left( \cos\left( e^{\left( 3^S + \frac{4e^{-S}}{5/3} \right)} \right) \right) \right) \right)} \right) - [0, 2].$$

*The package InCLosure guarantees arbitrarily sharper intervals which are restricted only by the machine's computational capabilities. The $\Delta_{\mathcal{J}}$-pair $(q, \delta q)_{[-1,2]}$ for the $\mathcal{J}$-function $q$ at $[-1, 2]$ can be computed using the following InCL command.*

```
IAD "[-0.5,-0.4]*X^10-[-3,-2]*X^6-sin(e^(5/8*(sin(cos(e^(3^X+e^-X/5/3*4))))))-[0,2]"
1 "X=[-1,2]"
```

*This will result in*

$$\left( \begin{array}{c} [-515.0, 191.4427985505345723919], \\ [-147294.44809204868892190371, 145297.44809204868892190371] \end{array} \right).$$

*The second parameter, "1", in the preceding InCL command, is the number of subdivisions (which means no subdivisions). To get sharper results, one can increase the number of subdivisions arbitrarily.* Table 1 *shows InCLosure values for the $\Delta_{\mathcal{J}}$-pair $(q, \delta q)_{[-1,2]}$ computed by subdividing $[-1, 2]$ into 1, 5, 10, 20, 50, and 100 subintervals.*

Example 9.4 shows that computing auto-derivatives under interval uncertainty, with the *arbitrarily* sharper and guaranteed results of InCLosure, is competitive and obviously preferable to the ordinary numerical approximation methods.

In closing, let us get a grip on how to compute *higher-order* interval auto-derivatives in the framework of our theory $\mathsf{T}_{\Delta_{\mathcal{J}}}$ of *dyadic* $\Delta_{\mathcal{J}}$-numbers ($\Delta_{\mathcal{J}}$-pairs). By virtue of

*Leibniz's rules for* $\Delta_{\mathcal{J}}$-*numbers* (theorem 6.2), and once we have extended the class $\mathbf{P}_{\langle 1 \rangle}$ of $\Delta_{\mathcal{J}}$-*kernels* by including the higher order dyads $(u, \delta u)$, $(\delta u, \delta^2 u)$, ..., $(\delta^n u, \delta^{n+1} u)$, for an arbitrary $n$, we can compute *higher order* interval auto-derivatives by doing only *dyadic* $\Delta_{\mathcal{J}}$-arithmetic. Consequently, within our own development, one can implement higher order interval auto-differentiation without resorting to defining any sort of $n$-dimensional *Grassmann algebra* for $n$-ary vectors of the form $(u, \delta u, \ldots, \delta^n u)$. With this in mind, let us consider the $\mathcal{J}$-function

$$q(S) = \cos\left(S^2\right) + \ln(S).$$

We desire to compute the $\Delta_{\mathcal{J}}$-pairs $\left(\delta^1 q, \delta^2 q\right)$ and $\left(\delta^2 q, \delta^3 q\right)$ at some $S_0 \in \mathcal{J}$. Then, the class $\mathbf{K}_{\langle 1 \rangle}$ of *unary* $\mathcal{J}$-*kernels* should include $\Delta_{\mathcal{J}}$-pairs up to the *third order*. That is, for $q_1(S) = \cos(S)$, $q_2(S) = S^2$, and $q_3(S) = \ln(S)$, we should have respectively the following $\Delta_{\mathcal{J}}$-*kernels*

$$(\cos(S), -\sin(S)), (-\sin(S), -\cos(S)), (-\cos(S), \sin(S));$$

$$\left(S^2, 2S\right), (2S, 2), (2, 0);$$

$$\left(\ln(S), \frac{1}{S}\right), \left(\frac{1}{S}, -\frac{1}{S^2}\right), \left(-\frac{1}{S^2}, \frac{2}{S^3}\right).$$

Now to compute $\delta\left(q, \delta q\right) = \left(\delta q, \delta^2 q\right)$ at $S_0$, we have the following dyadic $\Delta_{\mathcal{J}}$-pairs for respectively $\cos\left(S^2\right)$ and $\ln(S)$

$$\left(\delta\left(q_1\left(q_2\right)\right), \delta^2\left(q_1\left(q_2\right)\right)\right)_{S_0}$$
$$= \begin{pmatrix} \delta q_1\left(q_2\left(S_0\right)\right) \times \delta q_2\left(S_0\right), \\ \delta^2 q_1\left(q_2\left(S_0\right)\right) \times \left(\delta q_2\left(S_0\right)\right)^2 + \delta q_1\left(q_2\left(S_0\right)\right) \times \delta^2 q_2\left(S_0\right) \end{pmatrix}$$
$$= \left(-2S_0 \sin\left(S_0^2\right), -4S_0^2 \cos\left(S_0^2\right) - 2\sin\left(S_0^2\right)\right),$$
$$\left(\delta q_3, \delta^2 q_3\right)_{S_0} = \left(\frac{1}{S_0}, -\frac{1}{S_0^2}\right),$$

where all the values in the above $\Delta_{\mathcal{J}}$-pairs are computed by direct evaluation of the $\Delta_{\mathcal{J}}$-*kernels*. Having now the required $\Delta_{\mathcal{J}}$-numbers for $\cos\left(S^2\right)$ and $\ln(S)$, one can simply add the resultant $\Delta_{\mathcal{J}}$-pairs by $\Delta_{\mathcal{J}}$-addition to get $\left(\delta q, \delta^2 q\right)$ at $S_0$.

Likewise, to compute the dyad $\left(\delta^2 q, \delta^3 q\right)$, we have the following dyadic $\Delta_{\mathcal{J}}$-numbers for respectively $\cos\left(S^2\right)$ and $\ln(S)$

$$\left(\delta^2\left(q_1\left(q_2\right)\right), \delta^3\left(q_1\left(q_2\right)\right)\right)_{S_0}$$
$$= \begin{pmatrix} \delta^2 q_1\left(q_2\left(S_0\right)\right) \times \left(\delta q_2\left(S_0\right)\right)^2 + \delta q_1\left(q_2\left(S_0\right)\right) \times \delta^2 q_2\left(S_0\right), \\ \left(\delta^3 q_1\left(q_2\left(S_0\right)\right) \times \left(\delta q_2\left(S_0\right)\right)^3 + 2\delta q_2\left(S_0\right) \times \delta^2 q_2\left(S_0\right) \times \delta^2 q_1\left(q_2\left(S_0\right)\right)\right) \\ + \left(\delta^2 q_1\left(q_2\left(S_0\right)\right) \times \delta q_2\left(S_0\right) \times \delta^2 q_2\left(S_0\right) + \delta q_1\left(q_2\left(S_0\right)\right) \times \delta^3 q_2\left(S_0\right)\right) \end{pmatrix}$$
$$= \left(-4S_0^2 \cos\left(S_0^2\right) - 2\sin\left(S_0^2\right), 8S_0^3 \sin\left(S_0^2\right) - 12S_0 \cos\left(S_0^2\right)\right),$$
$$\left(\delta^2 q_3, \delta^3 q_3\right)_{S_0} = \left(-\frac{1}{S_0^2}, \frac{2}{S_0^3}\right),$$

and by $\Delta_{\mathcal{J}}$-addition of the resultant $\Delta_{\mathcal{J}}$-numbers we get $\left(\delta^2 q, \delta^3 q\right)$ at $S_0$. This illustrates that the theory $\mathsf{T}_{\Delta_{\mathcal{J}}}$ of *dyadic* $\Delta_{\mathcal{J}}$-numbers ($\Delta_{\mathcal{J}}$-pairs) is completely sufficient for computing interval auto-derivatives of first and higher orders.

## CONCLUSION

As we detailed in the introduction and elsewhere, combining subtlety of ordinary automatic differentiation with reliability of interval mathematics results in an intervalized version of algorithmic differentiation, namely "interval differentiation arithmetic", which so markedly surpasses its ordinary counterpart in power and reliability. With the aid of interval mathematics, automatic differentiation can be intervalized to handle uncertainty in quantifiable properties of real world physical systems and accordingly provide the computational methods that suffice to deal with the important problem of "getting guaranteed bounds" of images of real functions and their derivatives; and so, this article has been devoted to recasting interval differentiation arithmetic in a formalized theory, by putting into a systematic form its fundamental notions, and thus attaining the advantage of a concrete algebraic foundation that has then enabled us to extend the theory in such a manner that adds to its power, reliability, and applicability.

In the first place, after formalizing some set-theoretical and logical notions of particular importance for our purpose, we gave an axiomatization of a theory of a differential interval algebra and then we presented the notion of an interval extension of a family of real functions, together with some analytic notions of interval functions. Secondly, we set up an axiomatic theory of interval differentiation arithmetic, as a two-sorted extension of the theory of a differential interval algebra, and then we gave the proofs for its categoricity and consistency. We consequently constructed the algebraic system of interval differentiation arithmetic, deduced its fundamental properties, and showed that it constitutes a multiplicatively non-associative $\mathsf{S}$-semiring in which multiplication is subalternative and flexible. Then, we established some monotonicity and isomorphism theorems for interval differentiation numbers and proved a result concerning the structure of real differentiation numbers. And, lastly, we gave a brief account of the computational implementation of interval differentiation arithmetic and showed how to concurrently compute guaranteed enclosures of images of both families of real functions and their first and higher order derivatives.

From the very beginning, our axiomatic system included the notion of an interval extension of a family of real functions and the differentiability criteria thereof. Also, our construction differs in that we did not make use of Clifford's dual numbers or Grassmann numbers that are repeatedly 'borrowed' or 'reinvented' in the literature as proposed algebraic characterizations respectively for first and higher-order algorithmic differentiation. Moreover, a well-known fact of logic is that a "categorical" formalization of a theory is the "best" characterization thereof. By dint of being *categorical*, the axiomatic theory presented in this work lays serious claim to being "best" in the sense that it rightly accounts, up to isomorphism, for *all structures* of interval differentiation arithmetic. Furthermore, along the course to the main business of this study, a number of useful notions

have been introduced and formalized within the context of the proposed theory. Among these, we can mention interval enclosure of a bounded set, interval extension of a real family, proper interval functions and the criteria thereof, differentiability and continuous differentiability of a real family, interval differentiability criteria, differential enclosure of a real family, differentiation extension of an interval function, and differentiability criterion for interval differentiation numbers.

We would also remark that, on the strength of our axiomatization, many nice consequences come for free: categoricity and consistency of both the theory of interval algebra and the theory of interval differentiation algebra follow immediately, criteria for differentiability of families of real functions and their interval extensions are easily established, and the algebras of intervals and real differentiation numbers are both isomorphically embedded in the algebra of interval differentiation numbers.

The main contribution is therefore both a "logico-algebraic formalization" and an "extension" of interval differentiation arithmetic. The article provides an axiomatization of a comprehensive algebraic theory of interval differentiation arithmetic based on clear and distinct elementary ideas of real and interval algebras. We extend this formalized theory in two directions. On the one hand, although we made use of neither Clifford's dual numbers nor Grassmann hyper-dual numbers, our new formalization of *dyadic* interval differentiation numbers fully addresses interval auto-derivatives of first and higher order. On the other hand, by virtue of introducing the notion of an interval extension of a family of real functions, the theory is extended to provide the mathematical tools to get guaranteed enclosures of the images of families of real functions and their derivatives. Noteworthy also is that with a few basic modifications, the categorical system axiomatized in this text can be extended analogously to compute *fuzzy auto-derivatives*. Nevertheless, despite all the aforementioned advantages, guaranteed interval enclosures come at a price: the interval subdivision method could be computationally inefficient when manipulating problems involving thousands of uncertain quantities. Fortunately, there are many ways out of this problems. Among these, we mention, without pretension to be complete, *Hansen's centered forms*, *remainder forms*, *Kulisch's complete intervals*, *Kaucher intervals*, and *Dawood's universal intervals* (For further details, see, *e.g.*, *Dawood & Dawood, 2019a*; *Dawood & Dawood, 2020*; *Dawood & Dawood, 2022*, and *Shary & Moradi, 2021*).

In conclusion, the "self-validating" feature of interval automatic differentiation makes it useful and applicable in a wide range of scientific fields. In engineering and physical sciences, a recurring problem is to compute the derivatives under parametric uncertainty. In this regard, an *intervalized* theory of algorithmic differentiation is believed to be very useful for manipulating problems involving quantifiable uncertainties. The system proposed in this article provides a rigorous and extended mathematical foundation for both real and interval automatic differentiation. Being both a formalization and an extension of differentiation arithmetic as it is currently practised, the authors believe that such a formalization, hopefully, might have a worthwhile impact on both theoretical research and real world applications, with computational advantages for the solutions of new types of practical problems which can be expressed in terms of the mathematical machinery presented in the body of this article.

## ACKNOWLEDGEMENTS

We would like to express our sincere thanks to the academic editor and the reviewers for their constructive and insightful comments that greatly contributed to improving this work.

### Funding

The authors received no funding for this work.

### Competing Interests

The authors declare there are no competing interests.

### Author Contributions

- Hend Dawood conceived and designed the experiments, performed the experiments, analyzed the data, performed the computation work, prepared figures and/or tables, authored or reviewed drafts of the article, conceptualization, formal analysis, and methodology, and approved the final draft.
- Nefertiti Megahed conceived and designed the experiments, performed the experiments, analyzed the data, authored or reviewed drafts of the article, formal analysis and methodology, and approved the final draft.

### Data Availability

To reproduce the results of the numerical examples of the 'Machine implementation of interval auto-differentiation' section, the latest version of InCLosure is available for download from CERN's archive at Zenodo: Hend Dawood. (2020). InCLosure (Interval enCLosure): A Language and Environment for Reliable Scientific Computing (3.0). Zenodo. https://doi.org/10.5281/zenodo.3903993.

An InCLosure input file and its corresponding output containing, respectively, the code and results of the examples are also available at Zenodo: Hend Dawood. (2023). InCLosure Code for Interval Automatic Differentiation [Data set]. Zenodo. https://doi.org/10.5281/zenodo.7595449.

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
