# Peer review of "Automatic differentiation of uncertainties: an interval computational differentiation for first and higher derivatives with implementation"

_PeerJ Computer Science, doi:10.7717/peerj-cs.1301_

## Round 0.1 · original submission · Minor Revisions

Based on the three referee reports I request a minor revision of the manuscript.

With your revision, please indicate how you have responded to the referees' suggestions.

Please ignore the suggestion of one referee that "I recommend authors to add a list of symbols used at the beginning or end of the manuscript".

·

Basic reporting

The manuscript presents research related to improving the effectiveness of reliable knowledge acquisition under conditions of uncertainty. To achieve this goal, the interval analysis is proposed, and for this purpose appropriate tools are created and various examples are considered. In this regard, the structure of the manuscript is appropriate, using a sufficient number of relevant references on the research topic. The manuscript is well laid out and written in a good style.

Experimental design

To achieve the research objectives, an appropriate methodology based on analytical expressions and mathematical models was used.

Validity of the findings

Certainly, the presented results are important for engineering sciences, because most data are characterized by uncertainty, and in this regard, these studies are useful and applicable in many fields. In this sense, the conclusion section should emphasize the importance and applicability of the results in practice.

Additional comments

Overall, the work is well written and structured and my overall rating is positive. My main observations and comments are as follows:
- as many symbols and abbreviations are used in the text, I recommend authors to add a list of symbols used at the beginning or end of the manuscript.
- it would be good to improve the quality of all figures and graphics;
- it would be useful, in addition to the advantages in the conclusion section, for the authors to comment on the disadvantages of the proposed method. In this way, a complete picture of its qualities, capabilities and application limitations will be obtained.

Reviewer 2 ·

Basic reporting

The paper is very interesting, yet difficult for a non-mathematician.
While I consider myself an expert in interval computations, and am at least well familair with algorithmic differentiation, it was still hard for me (but interesting!).
Providing an axiomatic theory of differentiation over interval-valued domains is certainly interesting. I am not sure whether it is relevant from practival point of view. Will it affect AD codes somehow? Still the math was fascinating and cute.

Experimental design

The paper is mostly theoretical, but there is a software - InCLosure - related to it.
I find it really awkward that the software is available in binary form only, making it impossible to investigate its correctness. It is the first time I have met such a situation, in an academic research; moreover, in a rather theoretical one.
As far as I understand keeping closed-source some business-related projects, wouldn't it be a good idea to go open-source for a purely research, math-related code?

Validity of the findings

No comments.

Additional comments

Page 2: please mention also Norbert Wiener, and Mieczyslaw Warmus among the pioneers of the interval calculus: https://www.cs.utep.edu/interval-comp/early.html

Page 7: a D-number is defined, but no numbered definition is devoted to it. This is a pretty important notion, and we use it again only ten pages leter (page 17). This makes the `narration' even more difficult to follow.

Page 30: please remove the sentence `Having now reached the end of the article, it may not be amiss if we briefly consider our chief results, along with casting a glance ahead.' It sounds silly, at least IMHO.

Reviewer 3 ·

Basic reporting

In this paper, the authors has intended to lay out a systematic theory of dyadic interval differentiation numbers that wholly addresses first and higher order automatic derivatives under ncertainty.
The paper is well written. There is a nice flow of ideas arranged in a logical manner. The authors have gone through an extensive literature review in the introduction section that provide sufficient background knowledge of the subject discussed. Logical development of the new presented ideas with properly cited references make it a self contained document.

Experimental design

The authors has shown in detail process to computationally realize interval automatic differentiation. Many examples are given, illustrating automatic differentiation of interval functions and families of real functions.

Validity of the findings

The main contribution include
i) a “logico-algebraic formalization”
ii) an “extension” of interval differentiation arithmetic.

The article provides an axiomatization of a comprehensive algebraic theory of interval differentiation arithmetic based on clear and distinct elementary ideas of real and interval algebras. The authors have extended this formalized theory. New formalization of dyadic interval differentiation numbers fully addresses interval auto-derivatives of first and higher order.

Additional comments

Excellent work

---

## Round 0.2 · accepted · Accept

I'm happy with the response to all the referee comments and confirm that the manuscript is ready for publication.